# The Limits of Optimal Pricing in the Dark*

**Quinlan Dawkins**
Department of Computer Science
University of Virginia
qed4wg@virginia.edu

**Minbiao Han**
Department of Computer Science
University of Virginia
mh2ye@virginia.edu

**Haifeng Xu**
Department of Computer Science
University of Virginia
hx4ad@virginia.edu

## Abstract

A ubiquitous learning problem in today's digital market is, during repeated interactions between a *seller* and a *buyer*, how a seller can gradually learn optimal pricing decisions based on the buyer's past purchase responses. A fundamental challenge of learning in such a strategic setup is that the buyer will naturally have incentives to manipulate his responses in order to induce more favorable learning outcomes for him. To understand the limits of the seller's learning when facing such a strategic and possibly manipulative buyer, we study a natural yet powerful buyer manipulation strategy. That is, before the pricing game starts, the buyer simply commits to "imitate" a different value function by pretending to always react optimally according to this *imitative value function*.

We fully characterize the optimal imitative value function that the buyer should imitate as well as the resultant seller revenue and buyer surplus under this optimal buyer manipulation. Our characterizations reveal many useful insights about what happens at equilibrium. For example, a seller with concave production cost will obtain essentially 0 revenue at equilibrium whereas the revenue for a seller with convex production cost is the *Bregman divergence* of her cost function between no production and certain production. Finally, and importantly, we show that a more powerful class of pricing schemes does not necessarily increase, in fact, may be harmful to, the seller's revenue. Our results not only lead to an effective prescriptive way for buyers to manipulate learning algorithms but also shed lights on the limits of what a seller can really achieve when pricing in the dark.

## 1 Introduction

Pricing is a basic question in microeconomics [24] as well as a ubiquitous problem in today's digital markets [16]. In its textbook style setup, there are two agents: a buyer (he) and a seller (she). The seller produces $d$ types of *divisible* goods for sale and has production cost $c(\boldsymbol{x})$ for producing *bundle* $\boldsymbol{x} \in \mathbb{R}_+^d$ of the $d$ goods. Using a standard linear pricing scheme with unit price $p_i$ for goods $i \in [d]$, the seller charges the buyer $\boldsymbol{x} \cdot \boldsymbol{p}$ when he purchases bundle $\boldsymbol{x} \in \mathbb{R}_+^d$ under price vector $\boldsymbol{p} \in \mathbb{R}_+^d$. Naturally, the buyer's optimal bundle for purchase depends on his *value* function $v(\boldsymbol{x})$ about the products. We assume that, given price vector $\boldsymbol{p}$, the rational buyer will always pick the optimal bundle that maximizes his quasilinear utility $[v(\boldsymbol{x}) - \boldsymbol{x} \cdot \boldsymbol{p}]$. The key question of interest is how the seller can compute the revenue-maximizing optimal prices assuming rational buyer purchase behaviors.

---

*This work is supported by a Google Faculty Research Award and an NSF grant CCF-2132506.

The above basic problem can be easily solved if the seller has access to the buyer's value function $v(\boldsymbol{x})$. However, in practice, this value function is usually private and unknown to the seller. To address this challenge, there has been a rich line of recent research which looks to design the optimal pricing scheme against an unknown buyer. Some of these work have adopted learning-based approaches that aim to compute the profit maximizing price by interacting with the buyer and gleaning information about the buyer's utility function [3, 33, 32]. These are motivated by the wide spread of e-commerce today where the seller can repeatedly interact with the same buyer or buyers from the same population with similar preferences. For example, in *online advertising*, ad exchange platforms learn to price advertisers from past behaviors; in *online retailing*, retailers learn to price customers from their past purchase history; and in *crowdsourcing*, platforms learn to reward workers' efforts. Another line of works look to design dynamic pricing mechanisms that dynamically adjusts the price based on the observed past buyer purchase behaviors with the objective of maximizing the aggregated total revenue [4, 5, 26, 27, 38].

A key challenge of pricing against such unknown buyers, regardless through learning or dynamic pricing, is that the buyer may manipulate his responses in order to mislead the seller's algorithm and induce more favorable outcomes for themselves. This is a fundamental issue when learning from strategic data sources, and is particularly relevant in this pricing setup due to the strategic nature of the problem. In this case, the buyer controls the information content to the seller, therefore he naturally has incentives to utilize this advantage to gain more payoff by strategically misleading the seller's algorithms. Indeed, as observed in previous studies, online advertisers may strategically respond to an ad exchange platform's prices in order to induce a lower future price [22]; Consumers strategically time their purchases in order to obtain lower prices at online retailing platforms [23].

To understand the limits of the seller's optimal pricing against such a strategic and possibly manipulative buyer, we put forward a natural model which augments the above basic pricing problem with only one additional step — i.e., we assume that the game starts with the buyer *committing* to a different value functions, coined the *imitative value function*, after which the seller will compute the optimal pricing scheme against this imitative buyer value function. The buyer's commitment to an imitative value function captures a simple yet powerful buyer manipulation strategy that can be used against any seller learning algorithms — that is, the buyer simply behaves consistently according to this imitative value function during the entire interaction process. Intuitively, the motivation of such commitment assumption comes directly from the fact that the seller has no information about the buyer and has to optimize pricing "in the dark". Consequently, if the buyer consistently behaves according to some different imitative value function, the seller is not able to distinguish this buyer from another buyer who truly has the value function (see more justifications of the commitment assumption in Section 3). Moreover, such an imitation strategy is also easy to execute in practice by the buyer regardless whether the seller is learning from him or is adopting dynamic pricing schemes. In fact, the buyer could even just report his imitative value function to the seller directly at the beginning of any interaction. In this situation, no learning or dynamic pricing will be needed as the buyer will indeed always behave according to the imitative value function. Therefore, all the seller can do is to directly apply the optimal pricing scheme for the buyer's imitative value function.

We remark that such imitation-based manipulation strategy has attracted much interest in recent works, with similar motivations as us — i.e., trying to understand how to manipulate learning algorithm or conversely, how to design strategy-aware learning algorithms to mitigate such manipulation. However, most of these works have focused on the general Stackelberg game model [19, 10] as well as the Stackelberg security games [18, 30, 29]. Our optimal pricing problem is also a Stackelberg model and thus a natural fit for the study. The crucial difference between our work and previous studies is that both agents in our model have *continuous* utility functions whereas all these previous works [19, 18, 30, 10] have discrete agent utility functions. Therefore, our model leads to a *functional* analysis and optimization problem, which is more involved. Fortunately, we show that the optimal functional solution can still be characterized by leveraging the structure of the pricing problem.

## 1.1 Our Results and Implications

Given the effectiveness and easy applicability of the buyer's imitative strategy described above, this paper studies what the optimal buyer imitative value function is and how it would affect buyer's surplus and the seller's revenue. Our main result provides a full characterization about the optimal buyer imitative value function. We show that the optimal buyer imitative value function $u^*$ features a specific bundle of products $\boldsymbol{x}^*$ that is most desirable to the buyer. Interestingly, it turns out

that $u^*$ is a Leontief-type piece-wise linear concave value function [2] such that the buyer would only proportionally value a fraction of the desired bundle $x^*$ and nothing else. Moreover, we also characterize the seller revenue and buyer surplus at equilibrium as well as necessary and sufficient conditions under which the seller obtains strictly positive revenue.

The optimal buyer imitative value function turns out to depend crucially on the seller's production cost function. When the cost function is concave, we show that the optimal buyer imitative value function $u^*$ will "squeeze" the seller revenue essentially to $0$. The fortunate news for the seller, however, is that finding out the $u^*$ turns out to be an NP-hard task for the buyer. In fact, we prove that it is NP-hard to find the $u^*$ that can guarantee a polynomial faction of the optimal buyer surplus. For the widely adopted *convex* production cost, we show that the equilibrium seller revenue is the Bregman divergence between production $\mathbf{0}$ and the $x^*$ bundle mentioned above. This illustrates an interesting message that convex production costs are "better" at handling buyer's strategic manipulations. Note that production costs are indeed more often believed to be convex since economic models typically assume that marginal costs increase as quantity goes up [35, 34].

All our characterizations so far focus on the standard *linear* pricing scheme. Our last result examines the possibility of using a more general class of pricing schemes to address the buyer's strategic manipulation. Surprisingly, we show that for the strictly more general class of concave pricing functions (i.e., the seller is allowed to use any concave function as a pricing function), the equilibrium will remain the same as described above. Therefore, the more general pricing schemes do not necessarily help to address buyer's manipulation behavior. In fact, we show that there exist examples where strictly *broader* class of pricing functions leads to strictly worse seller revenue. This is because more general pricing schemes may "overfit" buyer's incentives, which renders it easier to manipulate. This illustrates an interesting phenomenon in learning from strategic data sources and shares similar spirit to *overfitting* in standard machine learning tasks.

## 1.2 Additional Related Works

Due to space limit, here we only briefly discuss the most related works while refer readers to Appendix A.1 for more detailed discussions and comparisons. Closely related to ours is a recent study by Tang and Zeng [36]. They study the bidders' problem of committing to "imitate" a fake value *distribution* in auctions and acting consistently as if the bidder's value were from the fake distribution. This is similar in spirit to our buyer's commitment to an imitative value *function*. However, there is significant difference betweens our setting and that of [36], which leads to very different conclusions as well. Specifically, the seller in [36] auctions a single indivisible item with no production costs whereas our seller sells multiple divisible items with production costs. Another recent work [28] also studies buyer's strategic manipulation against seller's pricing algorithms, but in a single-item multi-buyer situation. Moreover, they assume a fixed seller learning strategy (thus not adaptive to buyer's strategy) with separated exploration-exploitation phases motivated by [4]. Another very relevant literature is learning the optimal prices or optimizing aggregated total revenue by repeatedly interacting with a single buyer [4, 5, 26, 27, 38]. These works all focus on designing learning algorithms that can learn from strategically buyer responses. Our work complements this literature by studying the limits of what learning algorithms can achieve. Moreover, the setups of these previous works are also different from us – they either assume buyer values are drawn from distributions [4, 5, 27] or the seller sells a single indivisible good with discrete agent utilities. Thus, their results are not comparable to us.

There have also been studies on learning the optimal prices from *truthful* revealed preferences, i.e., assuming the buyer will honestly best respond to seller prices [8, 39, 6, 42, 3, 33, 32]. Our works try to understand if the assumption of truthful revealed preferences does not hold and if the buyer will strategically respond to the seller's learning, what learning outcome could be expected when the buyer simply imitates a different value function that is optimally chosen. From this perspective, these works serve as a key motivation for the present paper. More generally, our work subscribes to the general line of research on learning from strategic data sources. Most works in this space has focused on classification [11, 20, 43, 17, 25, 21, 12], regression problems [31, 15, 13] and distinguishing distributions [40, 41]. Our work however focuses on learning the optimal pricing scheme.

## 2 Preliminaries

**Basic Setup of the Optimal Pricing Problem.**  A seller (*she*) would like to sell $d$ different types of *divisible* goods to a buyer (*he*). It costs her $c(x)$ to produce $x \in \mathbb{R}^d$ bundle of these goods. Let

$X \subset \mathbb{R}^d$ denote the set of all feasible bundles that the seller can produce. We assume $X$ is convex, closed and has positive measure. As a standard assumption [24, 33, 8, 6, 39], the buyer has a *concave* value function $v(\boldsymbol{x})$ for any goods bundle $\boldsymbol{x} \in X$. We do not make any assumption about the seller's production cost $c(\boldsymbol{x})$, except that it is monotone non-decreasing. For normalization, we assume $\boldsymbol{0} \in X$ and $v(\boldsymbol{0}) = c(\boldsymbol{0}) = 0$.

The seller aims to find a revenue-maximizing pricing scheme assuming rational buyer behaviors. A seller pricing scheme is a function $p(\boldsymbol{x})$ that specifies the sale price for any bundle $\boldsymbol{x}$. By convention, $p(\boldsymbol{0}) = 0$ always. Let the set $\mathcal{P}$ denote the set of all pricing *functions* that are allowed to use by the buyer. The majority of this paper will focus on the textbook-style *linear* pricing scheme [24]. A linear pricing scheme is parameterized by a price vector $\boldsymbol{p}$ (to be designed) such that $p(\boldsymbol{x}) = \boldsymbol{p} \cdot \boldsymbol{x}$ where $i$'th entry $p_i$ is interpreted as the unit price for goods $i$. Let set

$$\mathcal{P}_L = \{p : p(\boldsymbol{x}) = \boldsymbol{p} \cdot \boldsymbol{x} \text{ for some } \boldsymbol{p} \in \mathbb{R}_+^d\}$$

denote the set of all linear pricing functions. In Section 6 we will also study the broader classes of *concave* pricing schemes where the set $\mathcal{P}$ consists of all monotone non-decreasing concave functions.

For any price function $p \in \mathcal{P}$, a rational buyer looks to purchase bundle $\boldsymbol{x}^*$ that maximizes his *utility*; That is, $\boldsymbol{x}^* = \arg\max_{\boldsymbol{x} \in X} [v(\boldsymbol{x}) - p(\boldsymbol{x})]$. Ties are broken in favor of the seller.[2] The buyer's best response can thus be viewed as a function $\boldsymbol{x}^*(p)$ of the seller's price function $p(\boldsymbol{x})$. Knowing that the buyer will best respond, the seller would like to pick the pricing function $p^* \in \mathcal{P}$ to maximize her revenue. Formally, $p^*$ is the solution to the following bi-level optimization:

$$p^* = \underset{p \in \mathcal{P}}{\operatorname{argmax}} [p(\boldsymbol{x}^*(p)) - c(\boldsymbol{x}^*(p))], \quad \text{where } \boldsymbol{x}^*(p) = \underset{\boldsymbol{x} \in X}{\operatorname{argmax}} [v(\boldsymbol{x}) - p(\boldsymbol{x})] \quad (1)$$

The optimal solution $(p^*, \boldsymbol{x}^*(p^*))$ to such a bi-level optimization problem forms an *equilibrium* of this pricing game. More formally, this is often called the *optimal* Stackelberg equilibrium or *strong* Stackelberg equilibrium [14, 33]. We call $p^*$ the *equilibrium pricing function* and $\boldsymbol{x}^*$ the *equilibrium bundle*. Note that this is a challenging *functional* optimization problem since the seller is picking a function $p \in \mathcal{P}$, while not a vector variable. However, when $\mathcal{P} = \mathcal{P}_L$ is the set of linear pricing scheme, the above problem becomes a bi-level variable optimization problem since any $p(\boldsymbol{x}) \in \mathcal{P}_L$ can be fully characterized by a price vector $\boldsymbol{p}$.

**Terminologies from Convex Analysis.** We defer basic definitions like convex/concave functions and super/sub-gradients to Appendix B, and only mention two useful notations here: (1) the set of super/sub-gradient for concave/convex function $f$ is denoted as $\partial f(\mathbf{x})$; (2) The *Bregman divergence* of a function $f$ is defined as $D_f(\mathbf{z}, \mathbf{x}) = f(\mathbf{z}) - f(\mathbf{x}) - \nabla f(\mathbf{x}) \cdot [\mathbf{z} - \mathbf{x}]$. $D_f(\mathbf{z}, \mathbf{x})$ is an important distance notion and is strictly positive for strictly convex functions when $\mathbf{z} \neq \mathbf{x}$.

## 3 A Model of Pricing Against a Deceptive Buyer in the Dark (PADD)

As mentioned in related work section 1.2, the literature of algorithms to learn pricing schemes from unknown buyers is massive. This work, however, takes a different perspective and seeks to understand how a buyer can strategically deceive the seller, through a simple yet effective class of manipulation strategies. Our model naturally captures a buyer's strategic responses to seller's pricing algorithms when the seller has no prior knowledge about the buyer, i.e., pricing "in the dark".

Thus, we study a buyer manipulation strategy that is *oblivious* to any pricing algorithm. That is, the buyer simply imitates a different value function $u(\boldsymbol{x})$ by consistently responding to *any* seller acts according to $u(\boldsymbol{x})$. Consequently, whatever the seller learns will be with respect to this *imitative value function* $u(\boldsymbol{x})$. Alternatively, one can think of the buyer as *committing* to always behave according to value function $u(\boldsymbol{x})$. Nevertheless, the buyer's objective is still to maximize his *true* utility by carefully crafting an imitative value function $u(\boldsymbol{x})$ to commit to. Given the buye's commitment to value function $u(\boldsymbol{x})$, the best pricing scheme for the seller is to use the optimal pricing function against buyer value function $u(\boldsymbol{x})$. Similar to $v(\boldsymbol{x})$, the imitative value functions $u(\boldsymbol{x})$ is assumed to be concave and monotone non-decreasing as well. Let set $\mathcal{C}$ denote the set of all such functions. These resulted in the following model of *Pricing Against a Deceptive buyer in the Dark* (PADD):

---

[2]This is usually without loss of generality since the seller can always induce desirable tie breaking by providing a negligible additional incentive to the buyer.

- The buyer with true value function $v(\boldsymbol{x})$ (unknown to seller) commits to react optimally according to an *imitative value function* $u(\boldsymbol{x}) \in \mathcal{C}$.
- The seller learns the buyer's imitative value function $u(\boldsymbol{x})$ and compute the optimal pricing function $p^* \in \mathcal{P}$ by solving bi-level Optimization Problem (1) w.r.t. $u(\boldsymbol{x})$.
- The buyer observes the seller's pricing function $p^*$, and then follows his commitment to react optimally w.r.t. to $u(\boldsymbol{x})$ by purchasing bundle $\boldsymbol{x}^* = \arg\max_{\boldsymbol{x} \in X}[u(\boldsymbol{x}) - p(\boldsymbol{x})]$.

We remark that such commitment to a fake value function is not uncommon in the literature; similar assumptions have been adopted in many recent works in, e.g., auctions [36], general Stackelberg games [19, 10] and security games [18, 30, 29]. The buyer's ability of making such a commitment fundamentally comes from the fact that the seller has no prior knowledge about the buyer's true value function $v(\boldsymbol{x})$, i.e., has to "price in the dark". We refer curious reader to Appendix A.2 for a more detailed discussion about this assumption.

Naturally, the buyer with true value function $v(\boldsymbol{x})$ would like to find the optimal imitative value function $u^*(\boldsymbol{x})$ to maximizes his utility. This results in the following equilibrium definition.

**Definition 1** (Equilibrium of PADD)**.** *The equilibrium of* PADD *consists of the optimal imitative value function $u^* \in \mathcal{C}$ that the buyer commits to, the seller's optimal pricing function $p^* \in \mathcal{P}$ against $u^*(\boldsymbol{x})$, and the buyer's response bundle $\boldsymbol{x}^* \in X$. Formally, $(u^*, p^*, \boldsymbol{x}^*)$ is an equilibrium for a buyer with true value function $v(\boldsymbol{x})$ to* PADD *if*

$$u^* = \underset{u \in \mathcal{C}}{\operatorname{argmax}}[v(\boldsymbol{x}^*) - p^*(\boldsymbol{x}^*)], \qquad \textit{where} \qquad p^* = \underset{p \in \mathcal{P}}{\operatorname{argmax}}[p(\boldsymbol{x}^*) - c(\boldsymbol{x}^*)],$$

$$\textit{where} \qquad \boldsymbol{x}^* = \underset{\boldsymbol{x} \in X}{\operatorname{argmax}}\big[u^*(\boldsymbol{x}) - p^*(\boldsymbol{x})\big]. \qquad (2)$$

The equilibrium of PADD gives rise to a challenging *tri-level functional optimization problem*.[3] Note that, the dependence of the buyer's objective $[v(\boldsymbol{x}^*) - p^*(\boldsymbol{x}^*)]$ on $u$ is indirectly through the two argmax problems afterwards. The buyer is assumed to know the production cost function $c(\boldsymbol{x})$, which is needed to compute his optimal $u^*$. This can be easily justified in situations where the seller has been on the market for some time, therefore her production cost gradually becomes public knowledge.

## 4 The Equilibrium of PADD under Linear Pricing, and Implications

In this section, we characterize the equilibrium of PADD under linear pricing schemes, i.e., $\mathcal{P} = \mathcal{P}_L = \{p : p(\boldsymbol{x}) = \boldsymbol{p} \cdot \boldsymbol{x} \text{ for some } \boldsymbol{p} \in \mathbb{R}_+^d\}$ consists of all linear pricing functions. A linear pricing function is determined by a non-negative price vector $\boldsymbol{p} \in \mathbb{R}_+^d$. To distinguish *functionals* from *vector variables*, we will use $P_L = \{\boldsymbol{p} : \boldsymbol{p} \in \mathbb{R}_+^d\}$ to denote the set of all possible non-negative price vectors so that each $\boldsymbol{p} \in P_L$ uniquely corresponds to a linear pricing function in $\mathcal{P}_L$. Under linear pricing, the equilibrium in Def. 1 denoted by $(u^*, \boldsymbol{p}^*, \boldsymbol{x}^*)$ is characterized by Eq. (2) where $\mathcal{P} = \mathcal{P}_L$.

Even with linear pricing, this is still a very challenging *tri-level functional* optimization problem since $u^*$ is a function chosen from the set of all possible functions from $X$ to $\mathbb{R}_+$, denoted by set $\mathcal{C}$. A first thought one might have is: why doesn't the buyer simply imitate $u^*(\boldsymbol{x}) = c(\boldsymbol{x})$. We will see later that this is not — in fact far from being — optimal since it makes the game zero-sum and the seller will pick a price that guarantee 0 revenue, e.g., a price of $\infty$. This leads the trade to happen at 0 production, which is clearly not optimal for the buyer. The optimal $u^*$ should provide some incentive for the seller to produce some amount $\boldsymbol{x}^*$ that is in some sense the best for the buyer. We will provide two concrete examples in the next section in Figure 2.

The main result of this section is the following characterization for the equilibrium of PADD. This general characterization does not depend on any specific property about function $v(\boldsymbol{x}), c(\boldsymbol{x})$.

**Theorem 1.** *In the equilibrium of* PADD *under linear pricing, the optimal buyer imitative value function $u^*$ can w.l.o.g. be written as the following concave function parameterized by production amount $\boldsymbol{x}^* \in X$ and a real value $p^* \in \mathbb{R}_+$:*

$$u^*(\boldsymbol{x}) = p^* \cdot \min\{\frac{x_1}{x_1^*}, \cdots \frac{x_d}{x_d^*}, 1\} \qquad (3)$$

---

[3]Even for linear pricing where $\mathcal{P} = \mathcal{P}_L$, this is still a functional optimization problem since the buyer's imitative value function $u$ is an arbitrary function in $\mathcal{C}$.

*where*

$$\boldsymbol{x}^* = \arg\max_{\boldsymbol{x}\in X}\left[v(\boldsymbol{x}) - \sup_{\alpha\in[0,1)}\frac{c(\boldsymbol{x}) - c(\alpha\boldsymbol{x})}{1-\alpha}\right] \quad and \quad p^* = \sup_{\alpha\in[0,1)}\frac{c(\boldsymbol{x}^*) - c(\alpha\boldsymbol{x}^*)}{1-\alpha} \quad (4)$$

*Moreover, under imitative value function $u^*(\boldsymbol{x})$,*

1. *For any vector $\lambda \in \Delta_d$ in the d-dimensional simplex, the linear pricing scheme with price vector $\boldsymbol{p}^* = (\lambda_1\frac{p^*}{x_1^*}, \lambda_2\frac{p^*}{x_2^*}, \cdots, \lambda_d\frac{p^*}{x_d^*})$ is optimal for $u^*$.*

2. *In any of the above optimal linear pricing schemes, the buyer's optimal bundle response is always $\boldsymbol{x}^*$ and the buyer payment will always equal $p^*$.*

3. *At equilibrium, the buyer surplus is $[v(\boldsymbol{x}^*) - p^*]$ and the seller revenue is $[p^* - c(\boldsymbol{x}^*)]$.*

Note that the $u^*$ described by Equation (3) may not be the unique optimal buyer imitative value function, but it is one of the optimal ones. Moreover, any optimal imitative value function will result in the same buyer surplus and seller revenue.

**Interpretation of Theorem 1.** Before a formal proof, it is worthwhile to take a closer look at the characterization of the buyer's optimal imitative value function $u^*$ characterized by Theorem 1 and the special pricing problem it ultimately induces. At a high level, the buyer has a desirable amount of products $\boldsymbol{x}^*$ in mind, characterized by the first equation in (4). He would pretend that his value for $\boldsymbol{x}^*$ equals $p^*$. Moreover, the value of any production amount $\boldsymbol{x}$ will linearly depend on the minimum possible coordinate-wise *fraction* of $\boldsymbol{x}^*$ that $\boldsymbol{x}$ contains, i.e., the $\min\{\frac{x_1}{x_1^*}, \cdots \frac{x_d}{x_d^*}, 1\}$ term in $u^*$.[4]

The $p^*$ is chosen as the largest possible *slope* of the segment between $c(\boldsymbol{x}^*)$ to $c(\alpha\boldsymbol{x}^*)$ among all possible $\alpha \in [0, 1)$ (see Figure 1 for an illustration in one dimension about how $p^*$ is chosen based on the cost function $c$). This is certainly a very carefully chosen value. Note that

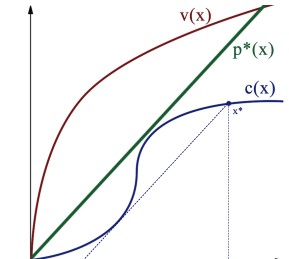

Figure 1: Illustration of $p^*$ in Theorem 1.

$$p^* = \sup_{\alpha\in[0,1)}\frac{c(\boldsymbol{x}^*) - c(\alpha\boldsymbol{x}^*)}{1-\alpha} \geq \frac{c(\boldsymbol{x}^*) - c(\boldsymbol{0})}{1-0} = c(\boldsymbol{x}^*),$$

which implies non-negativity of the seller's revenue $[p^* - c(\boldsymbol{x}^*)]$. In fact, the seller can achieve *strictly positive* revenue if and only if the above $\geq$ is strict. Finally, the optimal buyer imitative value function $u^*$ leads to a special pricing problem for the seller. Since the buyer's value is designed such that he is only interested in purchasing some fraction $r \in [0, 1]$ of $\boldsymbol{x}^*$, the seller's optimal pricing will charge $rp^*$ for the bundle $\boldsymbol{x} = r\boldsymbol{x}^*$ and this total charge $rp^*$ can be distributed *arbitrarily* over the $d$ products, e.g., by charging $\lambda_i rp^* = \lambda_i \frac{p^*\cdot x_i}{x_i^*}$ for product $i$ where $\sum_{i=1}^d \lambda_i = 1$. Overall, Theorem 1 fully characterized what the pricing problem is like at the equilibrium of PADD.

**Remark 1.** *Theorem 1 only provides a structural characterization about the equilibrium but does not imply that the equilibrium $(u^*, \boldsymbol{p}^*, \boldsymbol{x}^*)$ can be computed efficiently since we still need to solve the optimization problem (4) to find the optimal $\boldsymbol{x}^*$. As we will show later, this turns out actually to be an NP-hard problem in general. This is an interesting situation where the result reveals useful structural insights despite its computational intractability.*

*Proof Sketch of Theorem 1.* The proof of Theorem 1 is somewhat involved. We give a sketch here and defer formal arguments to Appendix C. The most challenging part is to find the optimal function $u^* \in \mathcal{C}$, which is a *functional optimization* problem. Standard optimization analysis only apply to programs with vector variables, while not functional variables. To overcome this challenge, our proof has two major steps. First, we argue that the concave functions of the specific format as in Equation (3) would suffice to help the buyer to achieve optimality. This effectively reduce the functional

---

[4]Notably, the format of this value function appears to have interesting connections to the well-known *Leontief production function* [2] which has the format of $\min_i\{\frac{x_i}{a_i}\}$. However, leontief functions are used to describe seller's production *quantities* as a function of quantities of different factors with no substitutability. It is an expected surprise that similar type of *value* function turns out to be optimal for buyer's strategic manipulation.

optimization problem to a variable optimization problem since any function of Format (3) can be characterized by $d + 1$ variables. Second, we then analyze the variable optimization problem we get and prove characterization of its optimal solutions. The first step is the most involved part and uses a significant amount of convex analysis. Such complication comes from the reasoning over the tri-level optimization problem (2). Tri-level optimization is generally highly intractable [9] — indeed, as we will show later, computing the equilibrium $(u^*, \boldsymbol{p}^*, \boldsymbol{x}^*)$ is NP-hard in general. Nevertheless, our analysis was able to bypass the difficulty by leveraging the special structure of the pricing problem and leads to a clean and useful characterization for the structure of the equilibrium.

A crucial intermediate step is the following characterization for a slightly simpler version of the question. That is, fixing any bundle $\overline{\boldsymbol{x}} \in X$, which imitative value function $u(\boldsymbol{x})$ will maximize the utility of the *buyer* with true value function $v(\boldsymbol{x})$, subject to that the optimal buyer purchase response under $u(\boldsymbol{x})$ is $\overline{\boldsymbol{x}}$? Fortunately, this question indeed admits a succinct characterization as shown below.

**Lemma 1.** *For any bundle $\overline{\boldsymbol{x}} \in X$, the optimal buyer imitative value function $\overline{u}(\boldsymbol{x})$, subject to that the resultant optimal buyer purchase response is bundle $\overline{\boldsymbol{x}}$, can without loss of generality have the following piece-wise linear concave function format, parameterized by a real number $\overline{p} \in \mathbb{R}$:*

$$\overline{u}(\boldsymbol{x}) = \overline{p} \cdot \min\{\frac{x_1}{\overline{x}_1}, \cdots \frac{x_d}{\overline{x}_d}, 1\} \tag{5}$$

*where $\overline{p} \in \mathbb{R}$ is the solution to the following linear program (LP):*

$$\begin{aligned} &\text{maximize} \quad v(\overline{\boldsymbol{x}}) - p \\ &\text{subject to} \quad p - c(\overline{\boldsymbol{x}}) \geq \alpha \cdot p - c(\alpha\overline{\boldsymbol{x}}), \quad \text{for } \alpha \in [0,1]. \end{aligned}$$  $\square$

## 5 Explicit Characterizations for Convex and Concave Costs

In this section, we instantiate Theorem 1 to both convex and concave cost functions, arguably the most widely adopted two classes of cost functions. In both cases, we give more explicit characterizations of the equilibrium outcome, including the buyer's optimal imitative value function as well as both agents' payoffs.

Convex production costs are widely adopted in many applications [7, 37]. When $c(\boldsymbol{x})$ is convex and differentiable, we show the following explicit characterization about the equilibrium outcome. A graphical visualization for this theorem is depicted in the left panel of Figure 2.

**Theorem 2.** *When $c$ is convex and differentiable, the piece-wise linear concave value function $u^*(\boldsymbol{x})$ defined by Equation (3), with $p^* = \boldsymbol{x}^* \cdot \nabla c(\boldsymbol{x}^*)$ and $\boldsymbol{x}^* = \arg\max_{\boldsymbol{x} \in X} [v(\boldsymbol{x}) - \boldsymbol{x} \cdot \nabla c(\boldsymbol{x})]$, is an optimal buyer imitative value function.*

*Under $u^*(\boldsymbol{x})$, the trade happens at bundle $\boldsymbol{x}^*$ with payment $p^* = \boldsymbol{x}^* \cdot \nabla c(\boldsymbol{x}^*)$. The seller revenue $[\boldsymbol{x}^* \cdot \nabla c(\boldsymbol{x}^*) - c(\boldsymbol{x}^*)]$ is precisely the Bregman divergence $D_c(\boldsymbol{0}, \boldsymbol{x}^*)$ between $\boldsymbol{0}$ and $\boldsymbol{x}^*$. The buyer surplus is $[v(\boldsymbol{x}^*) - \boldsymbol{x}^* \cdot \nabla c(\boldsymbol{x}^*)]$.*

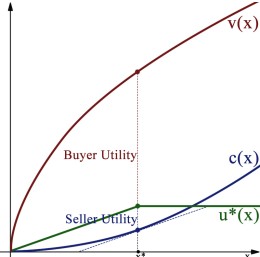
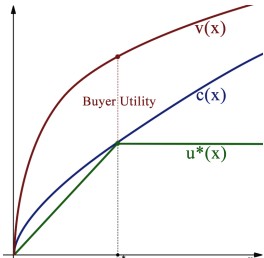

(a): An example for convex cost: $v(x) = 64\sqrt{x}$, $c(x) = x^2$. Theorem 2 implies $x^* = 4$, $p^* = \nabla c(x^*) = 8$, seller utility $D_c(0,4) = 16$ and buyer utility $v(x^*) - p^* \cdot x^* = 96$.

(b): An example of one-dimensional concave costs: $v(x) = 4x^{\frac{1}{4}}$, $c(x) = \sqrt{x}$. Theorem 3 implies $x^* = 16$, $p^* = \frac{c(x^*)}{x^*} = 1/4$, seller utility is 0 and buyer utility $v(x^*) - p^* \cdot x^* = 4$.

Figure 2: Illustration of the equilibrium characterization: $v(x)$ is buyer's true value function, $c(x)$ is the seller's cost function, and $u^*(x)$ is the optimal buyer imitative value function.

We now consider concave costs and prove the following characterization. A graphical visualization for the theorem is depicted in the right panel of Figure 2.

**Theorem 3.** *When $c(\boldsymbol{x})$ is concave, the piece-wise linear concave value function $u^*(\boldsymbol{x})$ defined by Equation (3), with $p^* = c(\boldsymbol{x}^*)$ and $\boldsymbol{x}^* = \arg\max_{\boldsymbol{x} \in X} [v(\boldsymbol{x}) - c(\boldsymbol{x})]$, is an optimal buyer imitative value function.*

*Under $u^*(\boldsymbol{x})$, the trade happens at bundle $\boldsymbol{x}^*$ with payment $p^* = c(\boldsymbol{x}^*)$. The seller revenue will be $0$. The buyer extracts the maximum possible surplus $\max_{\boldsymbol{x} \in X}[v(\boldsymbol{x}) - c(\boldsymbol{x})]$.*

Graphical visualizations for the above two theorems are depicted in Figure 2. Note that both examples in Figure 2 show the sub-optimality of imitating $u^*(x) = c(x)$ for the buyer. To see this, note $u^*(x) = c(x)$ makes the game zero-sum. To guarantee non-negative revenue, the seller must pick a price $p$ such that the line $p \cdot x$ is always above $u^*(x)$. A buyer imitating $u^*(x) = c(x)$ will end up purchasing a $0$ amount in both cases, and thus are sub-optimal for him.

An important conceptual message from Theorem 3 is that when the seller production cost is concave and known to the buyer, the buyer can always come up with an imitative value function which squeeze the seller's revenue to its extreme, i.e., $0$.[5] Under the seller's optimal imitative value function, the trade will happen at his most favorable bundle amount $\boldsymbol{x}^*$ and the seller pays just the cost function $c(\boldsymbol{x}^*)$ to the seller.

Theorem 3 is certainly bad news for the seller. However, it is a *descriptive* result and only shows it is possible for the buyer to achieve maximum possible surplus. Our next result brings somewhat good news to the seller. Specifically, we show it is NP-hard for the buyer to optimize, even approximately, his optimal surplus. The hardness holds even when the production cost function $c(\boldsymbol{x})$ is concave, in which case the surplus is characterized by Theorem 3. This shows that even though in theory the buyer can derive large surplus from strategic manipulation, even approximately figuring out such an optimal manipulation is impossible in general, unless P = NP.

**Theorem 4.** *[Intractability of Equilibrium] It is NP-hard to approximate the buyer equilibrium surplus in PADD games to be within ratio $1/d^{1-\epsilon}$ for any $\epsilon > 0$. This hardness result holds even when the production cost function $c(\boldsymbol{x})$ is concave and the buyer's true value function $v(\boldsymbol{x})$ is simply the linear function $\sum_{i=1}^d x_i$.*

Formal proofs for both equilibrium characterizations and the hardness of approximating the buyer equilibrium surplus can be found in Appendix D. We note that an intriguing open question is whether optimal imitative value function can be computed for *convex* production cost $c(\boldsymbol{x})$. In this case, the optimal bundle $\boldsymbol{x}^*$ can be explicitly expressed as $\boldsymbol{x}^* = \operatorname{argmax}_{\boldsymbol{x}}[u(\boldsymbol{x}) - \boldsymbol{x} \cdot \nabla c(\boldsymbol{x})]$ for any value function $u$, however how to derive the optimal $u^*$ remains challenging.

## 6   The Risk of Over-exploiting Buyer's Incentives

To counteract the buyer's strategic manipulation, one of the most natural approaches is perhaps to use a richer or more powerful class of pricing schemes, as opposed to only using linear pricing. Such additional power of pricing will always increase the revenue when facing an honest buyer. Unfortunately, however, we show that it does not necessarily help in the presence of buyer manipulation — in fact, the seller may suffer the risk of *over-exploiting the buyer's incentives* so that it becomes easier for the buyer to manipulate. This phenomenon is similar in spirit to the *overfitting* phenomenon in machine learning. That is, using a richer hypothesis class does not necessarily reduce the testing error, though it does reduce the training error. We believe that our findings in this section partially explain why simple pricing schemes like linear pricing are preferred in reality.

We first prove that the strictly more general class of *concave* pricing schemes can never do better for the seller than the (much) restricted class of linear pricing. In fact, surprisingly, the equilibrium of PADD under concave pricing turns out to be exactly the same as the equilibrium under linear pricing. This time, our proof utilizes a crucial observation that under concave pricing, the tri-level optimization problem of PADD can be reduced to solving a bi-level optimization problem (in particular, FOP (6)). Recall that the proof of Theorem 1 also reduces the tri-level FOP to a bi-level FOP through a characterization about the price and optimal buyer bundle in Lemma 3. However, Lemma 3 does not hold any more if the seller uses the richer class of concave pricing schemes. Therefore, the FOP (6) we obtain here is different from the core FOP (11) we solve in the proof of Theorem 1. Nevertheless,

---

[5]In real practice, the buyer can slightly deviate from his value function to given a small $\epsilon$ amount of incentive for the seller to strictly prefer production.

through careful convex analysis, we are able to show that the optimal solution to FOP (6) also admits an optimal solution of similar structure as characterized by Theorem 1.

**Theorem 5.** *The equilibrium of* PADD *under* concave *pricing is exactly the same as the equilibrium under* linear *pricing as characterized by Theorem 1.*

*Proof Sketch of Theorem 5.* We start by examining how the use of concave pricing schemes may simplify FOP (2). Recall that both value functions and pricing functions are monotone non-decreasing and normalized to be 0 at $\mathbf{0}$. For any concave buyer value function $u \in \mathcal{C}$, it is easy to see that the optimal pricing function $p$ simply equals $u$ (i.e., charging buyer exactly his imitative value) and ask the buyer to break ties in favor of the seller by picking $\boldsymbol{x}^* = \arg\max_{\boldsymbol{x} \in X}[u^*(\boldsymbol{x}) - c(\boldsymbol{x})]$ to maximize the seller's revenue. Consequently, the use of concave pricing simplifies FOP 2 to the following bi-level functional optimization problem:

$$u^* = \underset{u \in \mathcal{C}}{\text{argmax}}[v(\boldsymbol{x}^*) - p^*(\boldsymbol{x}^*)], \text{ where } \boldsymbol{x}^* = \underset{\boldsymbol{x} \in X}{\text{argmax}}\left[u^*(\boldsymbol{x}) - c(\boldsymbol{x})\right]$$

We fix a particular bundle $\overline{\boldsymbol{x}}$ and examine what buyer value function $u \in \mathcal{C}$ would maximize the buyer's utility subject to that the trade will happen at bundle $\overline{\boldsymbol{x}}$. This results in the following *functional optimization problem* (FOP) for the buyer with *functional variable* $u$.

$$
\begin{aligned}
\text{maximize} \quad & v(\overline{\boldsymbol{x}}) - u(\overline{\boldsymbol{x}}) \\
\text{subject to} \quad & u(\overline{\boldsymbol{x}}) - c(\overline{\boldsymbol{x}}) \geq u(\boldsymbol{x}') - c(\boldsymbol{x}'), \quad \text{for } \boldsymbol{x}' \in X.
\end{aligned}
\tag{6}
$$

where the constraint means the seller's optimal price for value function $u$ is $u(\overline{\boldsymbol{x}})$ and thus the buyer best response amount is indeed $\overline{\boldsymbol{x}}$. The remaining proof relies primarily on the following lemma.

**Lemma 2.** *The following concave function is optimal to FOP* (6):

$$\overline{u}(\boldsymbol{x}) = \overline{p} \cdot \min\{\frac{x_1}{\overline{x}_1}, \cdots \frac{x_d}{\overline{x}_d}, 1\}, \quad \text{where } \overline{p} = \sup_{\alpha \in [0,1)} \frac{c(\overline{\boldsymbol{x}}) - c(\alpha\overline{\boldsymbol{x}})}{1 - \alpha}.
\tag{7}$$

Given this characterization, the buyer simply needs to look for the best $\overline{\boldsymbol{x}}$. This then leads to the same characterization as in Theorem (1) since Equation (7) is the same as the value function characterized in Equation (5). The proof of Lemma 2 is deferred to Appendix E.1. □

Theorem 5 shows that more general class of pricing schemes may not help the seller to obtain more revenue. One might then wonder whether it at least never hurts since if that is the case, at least it would never be a worse choice. Unfortunately, our following example shows that a richer class of pricing schemes may bring *strict harm* to the seller and *strict benefit* to the seller.

**Example 1** ( The Risk of Overexploiting Buyer Incentives)**.** *There is a single type of divisible good to sell, i.e., $d = 1$. Let the seller's production cost function be the convex function $c(x) = x^2$ and let the buyer's true value function $v(x)$ be the following piece-wise linear concave function*

$$v(x) = \begin{cases} 10x & 0 \leq x \leq 0.81 \\ 8.1 & x > 0.81 \end{cases}$$

Let $\mathcal{P}_C$ denote the set of all concave pricing schemes. The following proposition completes Example 1 and its proof can be found in Appendix E.2.

**Proposition 1.** *For the instance in Example 1, there exists pricing scheme class $\mathcal{P}$ with $\mathcal{P}_L \subset \mathcal{P} \subset \mathcal{P}_C$ such that when the seller changes from linear pricing class $\mathcal{P}_L$ to the richer class $\mathcal{P}$, the seller's revenue strictly* decreases *and the buyer's surplus strictly* increases *at the equilibrium of* PADD*.*

## 7 Conclusions

Motivated by optimal pricing against an unknown buyer, this paper put forwards a simple variant of the very basic pricing model by augmenting it with an additional stage of buyer commitment at the beginning. This motivation is driven by the seller's ignorance of the buyer's value function and thus have to price in the dark. We fully characterize the equilibrium of this new game model. The equilibrium reveals interesting insights about what the seller can learn, and how much seller revenue

and buyer surplus it may result in. We also show that more general class of pricing schemes may overfit the buyer's incentive and lead to a pricing game that is even easier for the buyer to manipulate.

Our results opens the possibilities for many other interesting questions. For example, given the risk of using a too general class of pricing schemes, what class of pricing schemes is a good compromise between extracting revenue and robust to buyer manipulations? Is linear pricing scheme the best such class or any other pricing scheme? Second, as the first study of our setup, we have chosen to focus on a simple setup where the seller has completely no knowledge about the buyer's value function. An interesting question is, how the seller's learning and resultant revenue may increase when the seller has some prior knowledge about the buyer's values. In fact, one natural modeling question is how to model the seller's prior knowledge about the buyer's value function. Is the prior knowledge about which subclass the value functions are from or about what distribution the parameters of the value functions are from? Finally, our model assumes that the buyer has full knowledge about the seller. An ambitious though extremely intriguing question to ask is what if the buyer also has incomplete knowledge about the seller and how to analyze the equilibrium under the information asymmetry from both sides.

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
