# A  Additional Discussions

## A.1  Additional Discussions on Related Works

Our work is related to a recent study by Tang and Zeng [36] who study the bidders' problem of committing to a fake type distribution in auctions and acting consistently as if the bidder's type were from the fake type distribution. This is similar in spirit to our buyer's commitment to an imitative value function. However, there are two key differences between our work and [36]. First, the seller in our model (realistically) has production cost where as the auction setting of [36] does not have production cost. This is an important difference because with $0$ production cost, the optimal manipulation in our case is trivial, i.e. the buyer will imitate a value function of $0$. However, this trivial solution does not arise in the model of [36] because in their setup there are multiple buyers (bidders) and the competition among bidders increases the auctioneer's revenue despite bidders' imitative or faking behaviors. This is the second key difference between our work and [36]. Therefore, our work illustrates how the production cost can affect the buyer's strategic manipulation and the seller's revenue whereas the work of Tang and Zeng [36] sheds light on how the bidders' competition affect the auctioneer's mechanism design and ultimate revenue. Though these two aspects are not comparable, we believe they are both interesting for a deep understanding.

Another very relevant literature is learning the optimal prices or optimizing aggregated total revenue by repeatedly interacting with a single buyer [4, 5, 26, 27, 38]. Similar to us, they also consider the buyer's strategic behavior that potentially tricks the seller's learning algorithm. However, these previous works all focus on designing learning algorithms that can handle strategic data sources. Our work can be viewed as a complement to this literature. Instead of proposing new algorithms, we focus on understanding the limits of what learning algorithms can achieve by analyzing a basic model which is a variant of the textbook-style optimal pricing model. Moreover, the setups of these previous works are also different from us, which make them not comparable to us. For example, some of these models [4, 5, 27] assume that the buyer's values for goods are drawn from distributions (a.k.a., demand distribution) and consequently his best responses to seller prices are stochastic with randomness inherited from his value distribution. However, our model assumes that the buyer has an unknown but fixed value function that drives his purchase responses. Such a response is the solution to buyer's optimization problem while not from a random distribution. There have also been models that consider unknown but fixed buyer values like us [26, 38]. However, these works have focused on a single indivisible good with discrete agent utilities whereas our model has *multiple divisible* goods with continuous agent utilities.

There have also been studies on learning the optimal prices from *truthful* revealed preferences, i.e., assuming the buyer will honestly best respond to seller prices [8, 39, 6, 42, 3, 33, 32]. Our works try to understand if the assumption of truthful revealed preferences does not hold and if the buyer will strategically respond to the seller's learning, what learning outcome could be expected when the buyer simply imitates a different value function that is optimally chosen. From this perspective, these works serve as a key motivation for the present paper.

More generally, our work subscribes to the general line of research on learning from strategic data sources. Strategic classification has been studied in other different settings or domains or for different purposes, including spam filtering [11], classification under incentive-compatibility constraints [43], online learning [17, 12], and understanding the social implications [1, 25, 21]. Finally, going beyond classification, strategic behaviors in machine learning has received significant recent attentions, including in regression problems [31, 15, 13], distinguishing distributions [40, 41].

## A.2  Additional Discussion on Buyer's Commitment

The buyer's ability of making such a commitment fundamentally comes from the fact that the seller has no prior knowledge about the buyer's true value function $v(\boldsymbol{x})$, i.e., has to "price in the dark".[6] To find a good pricing scheme, the seller may interact with the buyer to learn the his value function [39, 8, 6], or learn the optimal pricing scheme [33], or directly optimize the aggregated revenue

---

[6]By convention, all value functions are assumed to be concave. This can be equivalently viewed as a restriction to the buyer's manipulation imposed by the seller's (very limited) prior knowledge about concavity of buyer values. Later, we will also briefly discuss how the absence of this prior knowledge may lead to worst seller revenue.

during repeated interactions [4, 5, 26, 27, 38]. However, regardless what algorithm the seller may adopt, the buyer can always choose to consistently behave according to a carefully crafted different value function $u(\boldsymbol{x})$, i.e., the *commitment*. For example, suppose the seller tries to apply any machine learning algorithm, the buyer may respond by directly announcing his imitative value function $u(\boldsymbol{x})$ even before the learning starts and then behave consistently. In such scenarios, learning is even not needed since the best a seller can do is to respond with the optimal pricing against $u(\boldsymbol{x})$. Similarly, if the seller adopts any dynamic pricing mechanism, the buyer may respond similarly by announcing her value function $u(\boldsymbol{x})$. Since the seller lacks knowledge about the buyer's value, such imitative buyer behavior makes him indistinguishable from a buyer who truly has value function $u(\boldsymbol{x})$. Therefore, our equilibrium characterization in later sections will help to understand what the optimal imitative value function for the buyer is and what revenue the seller can possibly achieve when pricing against such a strategic buyer in the dark, i.e., without any prior knowledge.

We remark that though the imitative strategy may not always be the absolutely best possible strategy for the buyer, but it enjoys many advantages. First of all, as we will prove later, this strategy does lead to significant improvement to the buyer's utility[7] and, in fact, is provably the best possible (among all possible strategic behaviors that the buyer may adopt) in certain circumstances, e.g., when the seller's production cost function $c(\boldsymbol{x})$ is concave. Second, this imitative strategy is easy to adopt in practice and requires no knowledge about the seller pricing scheme. For example, this imitative strategy works equally well for any price learning algorithm so long as it can effectively learn the optimal price from the buyer. Third, it also has good long term effect since the seller cannot distinguish whether the buyer truly has value function $u(\boldsymbol{x})$ or not, and may just have to use the same learned price for future purchases from this buyer. However, we show later that any buyer behavior that is not consistently imitating a value function can be easily identified by the seller. In such situations, even though the seller ended up with some prices, she knows that it is not the truly optimal price for this buyer and may take this into account in future interactions.

## B  Technical Background: Concave/Convex Functions and Super/Sub-Gradients

Let $f(\boldsymbol{x}) : X \to \mathbb{R}$ be any function where $X \subset \mathbb{R}^d$ is the domain of $f$. A vector $\boldsymbol{p} \in \mathbb{R}^d$ is called a *super-gradient* for $f$ at $\boldsymbol{x} \in X$ if for any $\boldsymbol{z} \in X$ we have $f(\boldsymbol{z}) \leq f(\boldsymbol{x}) + \boldsymbol{p} \cdot (\boldsymbol{z} - \boldsymbol{x})$. Function $f$ is called *concave* if for any $\boldsymbol{x}, \boldsymbol{z} \in \mathbb{R}^d$ and any $\alpha \in [0, 1]$ we have $\alpha f(\boldsymbol{x}) + (1 - \alpha) f(\boldsymbol{z}) \leq f(\alpha \boldsymbol{x} + (1 - \alpha) \boldsymbol{z})$. Super-gradients do not always exist. However, a concave function has at least one super-gradient at any $\boldsymbol{x} \in X$. For a differentiable concave function $f$, its gradient $\nabla f(\boldsymbol{x})$ is the only super-gradient at $\boldsymbol{x}$ for any $\boldsymbol{x} \in X$. If $f$ is concave but not differentiable, it may have multiple super-gradients at some $\boldsymbol{x}$. In this case, we use $\partial f(\boldsymbol{x})$ to denote the *set* of all super-gradients of $f$ at $\boldsymbol{x}$. Among all super-gradients in $\partial f(\boldsymbol{x})$, of our particular interest is the following one: $\nabla_{\max} f(\boldsymbol{x}) = \operatorname{argmax}_{\boldsymbol{p} \in \partial f(\boldsymbol{x})} [\boldsymbol{x} \cdot \boldsymbol{p}]$. This is the super-gradient that maximize linear function $\boldsymbol{p} \cdot \boldsymbol{x}$. When $f$ is differentiable, $\nabla_{\max} f(\boldsymbol{x}) = \nabla f(\boldsymbol{x})$ is the (only) super-gradient.

Function $f$ is called *convex* if $-f$ is concave. For convenience of stating our results, we will mostly work with *differentiable* convex functions in this paper. A useful distance notion for differentiable convex function $f$ is the *Bregman divergence*:

$$D_f(\boldsymbol{z}, \boldsymbol{x}) = f(\boldsymbol{z}) - f(\boldsymbol{x}) - \nabla f(\boldsymbol{x}) \cdot [\boldsymbol{z} - \boldsymbol{x}].$$

$D_f(\boldsymbol{z}, \boldsymbol{x})$ is always non-negative for convex functions and strictly positive for strictly convex functions when $\boldsymbol{z} \neq \boldsymbol{x}$. However, Bregman divergence is asymmetric among variables, i.e., $D_f(\boldsymbol{z}, \boldsymbol{x}) \neq D_f(\boldsymbol{x}, \boldsymbol{z})$ in general.

## C  Proof of Theorem 1

**Theorem 1.** *In the equilibrium of* PADD *under linear pricing, the optimal buyer imitative value function $u^*$ can w.l.o.g. be written as the following concave function parameterized by production amount $\boldsymbol{x}^* \in X$ and a real value $p^* \in \mathbb{R}_+$:*

$$u^*(\boldsymbol{x}) = p^* \cdot \min\{\frac{x_1}{x_1^*}, \cdots \frac{x_d}{x_d^*}, 1\} \tag{3}$$

---

[7]It is never worse since the buyer can always at least behave truthfully by letting $u(\boldsymbol{x}) = v(\boldsymbol{x})$.

*where*

$$\boldsymbol{x}^* = \arg\max_{\boldsymbol{x} \in X} \Big[ v(\boldsymbol{x}) - \sup_{\alpha \in [0,1)} \frac{c(\boldsymbol{x}) - c(\alpha\boldsymbol{x})}{1-\alpha} \Big] \quad \text{and} \quad p^* = \sup_{\alpha \in [0,1)} \frac{c(\boldsymbol{x}^*) - c(\alpha\boldsymbol{x}^*)}{1-\alpha} \quad (4)$$

*Moreover, under imitative value function $u^*(\boldsymbol{x})$,*

1. *For any vector $\lambda \in \Delta_d$ in the d-dimensional simplex, the linear pricing scheme with price vector $\boldsymbol{p}^* = (\lambda_1 \frac{p^*}{x_1^*}, \lambda_2 \frac{p^*}{x_2^*}, \cdots, \lambda_d \frac{p^*}{x_d^*})$ is optimal for $u^*$.*

2. *In any of the above optimal linear pricing schemes, the buyer's optimal bundle response is always $\boldsymbol{x}^*$ and the buyer payment will always equal $p^*$.*

3. *At equilibrium, the buyer surplus is $[v(\boldsymbol{x}^*) - p^*]$ and the seller revenue is $[p^* - c(\boldsymbol{x}^*)]$.*

We start with a useful lemma that characterizes the relation between optimal seller price $\boldsymbol{p}_u$ and optimal buyer bundle $\boldsymbol{x}_u$ for any concave buyer utility function $u(\boldsymbol{x})$. A similar result has been proved in [33]. The only difference here is that we allow any concave buyer utility function whereas the buyer utility function in [33] is assumed to be strictly concave and differentiable. Nevertheless, the proof remains similar and thus is omitted due to space limit.

**Lemma 3.** *For any concave buyer value function $u(\boldsymbol{x}) : \mathbb{R}_+^d \to \mathbb{R}_+$ reported by the buyer, let $\boldsymbol{p}_u$ be the optimal price vector for the seller and $\boldsymbol{x}_u$ be the resultant buyer optimal bundle for purchase, then the following relation holds:*

$$\boldsymbol{p}_u = \nabla_{\max} u(\boldsymbol{x}_u). \tag{8}$$

*where $\nabla_{\max} u(\boldsymbol{x}) = \mathrm{argmax}_{\boldsymbol{p} \in \partial u(\boldsymbol{x})}[\boldsymbol{x} \cdot \boldsymbol{p}].$*[8]

A crucial intermediate step in our proof is the following characterization for a slightly simpler version of the question. That is, fixing any bundle $\overline{\boldsymbol{x}} \in X$, which imitative value function $u(\boldsymbol{x})$ will maximize the utility of the *buyer* with true value function $v(\boldsymbol{x})$, subject to that the optimal buyer purchase response under $u(\boldsymbol{x})$ is $\overline{\boldsymbol{x}}$? If we can find a succinct characterization for this simpler question, what remains is just to search for the best bundle $\overline{\boldsymbol{x}}$. That will be a (variable) optimization problem, which is suitable for standard optimization techniques to solve. Fortunately, the above question does admit a succinct characterization.

**Lemma 4.** *[Lemma 1 restated] For any bundle $\overline{\boldsymbol{x}} \in X$, the optimal buyer imitative value function $\overline{u}(\boldsymbol{x})$, subject to that the resultant optimal buyer purchase response is bundle $\overline{\boldsymbol{x}}$, can without loss of generality have the following piece-wise linear concave function format, parameterized by a real number $\overline{p} \in \mathbb{R}$:*

$$\overline{u}(\boldsymbol{x}) = \overline{p} \cdot \min\{\frac{x_1}{\overline{x}_1}, \cdots \frac{x_d}{\overline{x}_d}, 1\} \tag{9}$$

*where $\overline{p}$ is the solution to the following linear program (LP):*

$$\begin{array}{ll} \text{maximize} & v(\overline{\boldsymbol{x}}) - p \\ \text{subject to} & p - c(\overline{\boldsymbol{x}}) \geq \alpha \cdot p - c(\alpha\overline{\boldsymbol{x}}), \quad \text{for } \alpha \in [0,1]. \end{array} \tag{10}$$

*Moreover, under imitative value function $\overline{u}(\boldsymbol{x})$, we have*

1. *For any convex coefficients $\lambda \in \Delta_d$, the linear pricing scheme with unit price vector $(\lambda_1 \cdot \frac{\overline{p}}{\overline{x}_1}, \lambda_2 \cdot \frac{\overline{p}}{\overline{x}_2}, \cdots, \lambda_d \cdot \frac{\overline{p}}{\overline{x}_d})$ will be optimal.*

2. *In any of the above optimal linear pricing schemes, the buyer's optimal bundle response will always be $\overline{\boldsymbol{x}}$ and the buyer payment will equal $\overline{p}$.*

*Proof of Lemma 4.* From the buyer's perspective, with a fixed bundle $\overline{\boldsymbol{x}}$ in mind, his problem is to come up with an imitative value function $u(\boldsymbol{x})$ such that its corresponding price $\nabla_{\max} u(\overline{\boldsymbol{x}})$ as from Lemma 3 maximizes his revenue at bundle $\overline{\boldsymbol{x}}$. This results in the following *functional optimization problem* (FOP) for the buyer with *functional variable* $u$.

$$\begin{array}{ll} \text{maximize} & v(\overline{\boldsymbol{x}}) - \overline{\boldsymbol{x}} \cdot \nabla_{max} u(\overline{\boldsymbol{x}}) \\ \text{subject to} & \nabla_{max} u(\overline{\boldsymbol{x}}) \cdot \overline{\boldsymbol{x}} - c(\overline{\boldsymbol{x}}) \geq \nabla_{max} u(\boldsymbol{x}') \cdot \boldsymbol{x}' - c(\boldsymbol{x}'), \quad \text{for } \boldsymbol{x}' \in X. \\ & u \text{ is concave} \end{array} \tag{11}$$

---

[8]The "$\max$" comes from the fact that the seller will pick the profit-maximizing price if multiple prices result in the same optimal buyer purchase.

where the first constraint means the seller's optimal price for value function $u(\boldsymbol{x})$ is $\nabla_{max}u(\overline{\boldsymbol{x}})$ and thus the buyer best response bundle is indeed $\overline{\boldsymbol{x}}$.

The lemma states that the $\overline{u}(\boldsymbol{x})$ defined in Equation (9) is an optimal solution to FOP (11). To analyze FOP (11), we first simplify the class of concave function $u$ that we need to consider. In particular, we claim that there always exists an optimal solution to FOP (11) such that $u(\boldsymbol{x})$ has the following form:

$$u(\boldsymbol{x}) = p \cdot \min\{\frac{x_1}{\overline{x}_1}, \cdots \frac{x_d}{\overline{x}_d}, 1\} \tag{12}$$

where the only *parameter* $p \in \mathbb{R}$ is the coefficient vector. To prove this, let $u^*(\boldsymbol{x})$ be any optimal solution to FOP (11). Construct another concave value function $\overline{u}(\boldsymbol{x})$ as follows:

$$\overline{u}(\boldsymbol{x}) = \overline{p} \cdot \min\{\frac{x_1}{\overline{x}_1}, \cdots \frac{x_d}{\overline{x}_d}, 1\}, \text{ where } \overline{p} = \nabla_{\max}u^*(\overline{\boldsymbol{x}}) \cdot \overline{\boldsymbol{x}} \tag{13}$$

Note that $\overline{p}$ is precisely the payment for a buyer with value function $u^*$ when his optimal bundle amount is $\overline{\boldsymbol{x}}$. We show that this new value function will result in the same optimal buyer bundle $\overline{\boldsymbol{x}}$ and payment $\overline{p}$. It thus does not change either of the agent's utilities and remains optimal for the buyer.

First, we argue that the constructed $\overline{u}$ is still feasible to FOP (11). Concavity of $\overline{u}$ is evident since it is the minimum of a set of linear functions. By Lemma 3, we have $\nabla_{\max}\overline{u}(\boldsymbol{x}) = 0$ if $\boldsymbol{x}$ is element-wise strictly greater than $\overline{\boldsymbol{x}}$. Otherwise, let $i^* = \operatorname{argmin}_{\{i|x_i \leq \overline{x}_i\}} \frac{x_i}{\overline{x}_i}$,[9] we have $[\nabla_{\max}\overline{u}(\boldsymbol{x})]_{i^*} = \frac{\overline{p}}{\overline{x}_{i^*}}$ which is the $i^*$'th element of $\nabla_{\max}\overline{u}(\boldsymbol{x})$, while all the other gradient entries of $\nabla_{\max}\overline{u}(\boldsymbol{x})$ are 0. Therefore, we have $\nabla_{\max}\overline{u}(\boldsymbol{x}) \cdot \boldsymbol{x} = \overline{p} \cdot \frac{x_{i^*}}{\overline{x}_{i^*}}$. Specifically, $\nabla_{\max}\overline{u}(\overline{\boldsymbol{x}}) \cdot \overline{\boldsymbol{x}} = \overline{p}$ which equals precisely the buyer payment $\nabla_{\max}u^*(\overline{\boldsymbol{x}}) \cdot \overline{\boldsymbol{x}}$ under utility $u^*$.

We now verify that the constraints in FOP (11) still holds for $\overline{u}$. We start from verifying for special $\boldsymbol{x}'$s where there exists $\alpha \in [0,1]$ such that $\boldsymbol{x}' = \alpha \cdot \overline{\boldsymbol{x}}$. In this case, we have

$$
\begin{aligned}
\nabla_{\max}\overline{u}(\overline{\boldsymbol{x}}) \cdot \overline{\boldsymbol{x}} - c(\overline{\boldsymbol{x}}) &= \nabla_{\max}u^*(\overline{\boldsymbol{x}}) \cdot \overline{\boldsymbol{x}} - c(\overline{\boldsymbol{x}}) &&\text{(since } \nabla_{\max}\overline{u}(\overline{\boldsymbol{x}}) \cdot \overline{\boldsymbol{x}} = \overline{p} = \nabla_{\max}u^*(\overline{\boldsymbol{x}}) \cdot \overline{\boldsymbol{x}})\\
&\geq \nabla_{\max}u^*(\boldsymbol{x}') \cdot \boldsymbol{x}' - c(\boldsymbol{x}') &&\text{(by feasibility of } u^* )\\
&\geq \nabla_{\max}u^*(\overline{\boldsymbol{x}}) \cdot \boldsymbol{x}' - c(\boldsymbol{x}') &&\text{(by concavity of } u^*)\\
&= \alpha \cdot \overline{p} - c(\boldsymbol{x}') &&\text{(since } \boldsymbol{x}' = \alpha \cdot \overline{\boldsymbol{x}} \text{ and } \overline{p} = \nabla_{\max}u^*(\overline{\boldsymbol{x}}) \cdot \overline{\boldsymbol{x}})\\
&= \nabla_{\max}\overline{u}(\boldsymbol{x}') \cdot \boldsymbol{x}' - c(\boldsymbol{x}'). &&\text{(by definition of } \overline{u} \text{ and } \boldsymbol{x}' = \alpha \cdot \overline{\boldsymbol{x}})
\end{aligned}
$$

Specifically, the second inequality holds because $\nabla_{\max}u^*(\alpha\overline{\boldsymbol{x}}) \cdot \overline{\boldsymbol{x}}/|\overline{\boldsymbol{x}}|$ is the directional derivative of $u^*$ at $\alpha\overline{\boldsymbol{x}}$ in the direction of $\overline{\boldsymbol{x}}$, which is non-increasing with respect to $\alpha$ due to the concavity of $u^*$. The above argument also implies $\nabla_{\max}\overline{u}(\overline{\boldsymbol{x}}) \cdot \overline{\boldsymbol{x}} - c(\overline{\boldsymbol{x}}) \geq 0$ by instantiating $\boldsymbol{x}' = \boldsymbol{0}$.

Next, we consider the case when $\boldsymbol{x}' \neq \alpha \cdot \overline{\boldsymbol{x}}$ for $\alpha \in [0,1]$. There will be two possible situations to consider:

1. If $\boldsymbol{x}'$ is element-wise greater than $\overline{\boldsymbol{x}}$, then $\nabla_{\max}\overline{u}(\boldsymbol{x}') = 0$. Thus, we have $\nabla_{\max}\overline{u}(\overline{\boldsymbol{x}}) \cdot \overline{\boldsymbol{x}} - c(\overline{\boldsymbol{x}}) \geq 0 > \nabla_{\max}\overline{u}(\boldsymbol{x}') \cdot \boldsymbol{x}' - c(\boldsymbol{x}') = 0 - c(\boldsymbol{x}')$.

2. If $\boldsymbol{x}'$ is *not* element-wise greater than $\overline{\boldsymbol{x}}$, let $i^* = \operatorname{argmin}_{\{i|x'_i \leq \overline{x}_i\}} \frac{x'_i}{\overline{x}_i}$ and $\alpha^* = \min_{\{i|x'_i \leq \overline{x}_i\}} \frac{x'_i}{\overline{x}_i}$. Then we have $[\nabla_{\max}\overline{u}(\boldsymbol{x}')]_{i^*} = \frac{\overline{p}}{\overline{x}_{i^*}}$ which is the $i^*$'th element of $\nabla_{\max}\overline{u}(\boldsymbol{x}')$, while all the other elements of $\nabla_{\max}\overline{u}(\boldsymbol{x}')$ are 0. In addition, denote $\widehat{\boldsymbol{x}}' = \alpha^* \cdot \overline{\boldsymbol{x}}$. Note that for $\boldsymbol{x}'$, we have $\nabla_{\max}\overline{u}(\boldsymbol{x}') \cdot \boldsymbol{x}' = \alpha^* \cdot \overline{p} = \nabla_{\max}\overline{u}(\widehat{\boldsymbol{x}}') \cdot \widehat{\boldsymbol{x}}'$. However, we have $c(\widehat{\boldsymbol{x}}') \leq c(\boldsymbol{x}')$ because $\widehat{\boldsymbol{x}}'$ is element-wise less than or equal to $\boldsymbol{x}'$. Thus, we have $\nabla_{\max}\overline{u}(\widehat{\boldsymbol{x}}') \cdot \widehat{\boldsymbol{x}}' - c(\widehat{\boldsymbol{x}}') \geq \nabla_{\max}\overline{u}(\boldsymbol{x}') \cdot \boldsymbol{x}' - c(\boldsymbol{x}')$ by monotonicity of $c(\boldsymbol{x})$. Our previous derivation for the special case $\boldsymbol{x}' = \alpha\overline{\boldsymbol{x}}$ with $\alpha \in [0,1]$ implies $\nabla_{\max}\overline{u}(\overline{\boldsymbol{x}}) \cdot \overline{\boldsymbol{x}} - c(\overline{\boldsymbol{x}}) \geq \nabla_{\max}\overline{u}(\alpha \cdot \overline{\boldsymbol{x}}) \cdot (\alpha \cdot \overline{\boldsymbol{x}}) - c(\alpha \cdot \overline{\boldsymbol{x}}), \forall \alpha \in [0,1]$. These together imply

$$\nabla_{\max}\overline{u}(\overline{\boldsymbol{x}}) \cdot \overline{\boldsymbol{x}} - c(\overline{\boldsymbol{x}}) \geq \nabla_{\max}\overline{u}(\widehat{\boldsymbol{x}}') \cdot \widehat{\boldsymbol{x}}' - c(\widehat{\boldsymbol{x}}') \geq \nabla_{\max}\overline{u}(\boldsymbol{x}') \cdot \boldsymbol{x}' - c(\boldsymbol{x}').$$

---

[9]If there are multiple $i$ that all minimize $\frac{x_i}{\overline{x}_i}$, the proof is valid by picking any of them.

As a result, the constructed $\overline{u}(\boldsymbol{x})$ is feasible to FOP (11) because $\nabla_{max}u(\overline{\boldsymbol{x}})\cdot\overline{\boldsymbol{x}}-c(\overline{\boldsymbol{x}})\geq\nabla_{max}u(\boldsymbol{x}')\cdot\boldsymbol{x}'-c(\boldsymbol{x}')$ for any $\boldsymbol{x}'\in X$.

Next, we argue that $\overline{u}(\boldsymbol{x})$ achieves the same buyer utility, and thus must also be optimal. This is simply because the feasibility of $\overline{u}(\boldsymbol{x})$ implies that the optimal buyer bundle will still be $\overline{\boldsymbol{x}}$ and payment will still be $\overline{p}=\nabla_{\max}u^*(\overline{\boldsymbol{x}})\cdot\overline{\boldsymbol{x}}$. As a result, buyer achieves the same utility when using $\overline{u}(\boldsymbol{x})$ and $u^*(\boldsymbol{x})$, yielding the optimality of $\overline{u}(\boldsymbol{x})$.

So far we showed that there always exists an optimal $u(\boldsymbol{x})$ of Form (9). Therefore, to solve FOP (11), we can without loss of generality focus on functions of the Form (9), which is parameterized $p$. By plugging in Form (9) into FOP (11), we obtain the following LP with variable $p\in\mathbb{R}$.

$$
\begin{aligned}
\text{maximize}\quad & v(\overline{\boldsymbol{x}})-p \\
\text{subject to}\quad & p-c(\overline{\boldsymbol{x}})\geq\alpha\cdot p-c(\boldsymbol{x}'), && \text{for } \boldsymbol{x}'=\alpha\cdot\overline{\boldsymbol{x}},\alpha\in[0,1]. \\
& p-c(\overline{\boldsymbol{x}})\geq 0-c(\boldsymbol{x}'), && \text{for } \boldsymbol{x}' \text{ element-wise greater than } \overline{\boldsymbol{x}}. \\
& p-c(\overline{\boldsymbol{x}})\geq\min_{\{i|x_i'\leq\overline{x}_i\}}\frac{x_i'}{\overline{x}_i}\cdot p-c(\boldsymbol{x}'), && \text{for other } \boldsymbol{x}'\in X. \\
& p\geq 0
\end{aligned}
$$

(14)

We now further simplify the above LP to become LP (10). That is, we argue that only the first constraint is needed and thus the other constraints can be omitted. When the first constraint is instantiated with $\boldsymbol{x}'=\boldsymbol{0}$, it implies $p-c(\overline{\boldsymbol{x}})\geq 0$. This immediately implies the *second* and the *last* constraint. By the proof above, we know that for any $\boldsymbol{x}'\neq\alpha\cdot\overline{\boldsymbol{x}}$ and $\boldsymbol{x}'$ is not element-wise greater than $\overline{\boldsymbol{x}}$ either, there must exist $\widehat{\boldsymbol{x}}'=\alpha^*\cdot\overline{\boldsymbol{x}}$ $(\alpha^*\in[0,1])$ such that $\nabla_{\max}\overline{u}(\widehat{\boldsymbol{x}}')\cdot\widehat{\boldsymbol{x}}'-c(\widehat{\boldsymbol{x}}')\geq\nabla_{\max}\overline{u}(\boldsymbol{x}')\cdot\boldsymbol{x}'-c(\boldsymbol{x}')$. Therefore, the *third* constraint is guaranteed to be satisfied as long as the first constraint is satisfied. As a result, the above LP can be further simplified to LP (10).

The constraint of LP (10) guarantee that the seller will maximize revenue at bundle $\overline{x}$. Since at $x=\overline{x}$, any $i$ will minimize the term $\frac{x_i}{\overline{x}_i}$. Lemma 3 then implies the optimal prices at $\overline{x}$ can be any convex combination of pricing vectors $(0,\cdots,0,\frac{\overline{p}}{\overline{x}_i},0,\cdots,0),\forall i$. The total payment will always be $\overline{p}$ under any of these optimal pricing schemes. This completes the proof. $\qquad\square$

Theorem (1) then follows from Lemma 4. We first observe that constraint in linear program (10) can be re-written as $p\geq\frac{c(\overline{\boldsymbol{x}})-c(\alpha\overline{\boldsymbol{x}})}{1-\alpha}$ for $\alpha\in[0,1)$ (the constraint is trivial for $\alpha=1$). Since the objective of LP (10) is equivalent to minimizing $p$, we thus have the optimal $p$ equals $\max_{\alpha\in[0,1)}\left[\frac{c(\overline{\boldsymbol{x}})-c(\alpha\overline{\boldsymbol{x}})}{1-\alpha}\right]$.

We have now characterized the optimal imitative value function for any fixed $\overline{\boldsymbol{x}}$. To compute the globally optimal imitative value function $u^*$, we only need to pick the $\overline{\boldsymbol{x}}$ that maximizes the buyer's surplus. By viewing $\overline{\boldsymbol{x}}$ as a variable $\boldsymbol{x}$, we obtain the desired form of $\boldsymbol{x}^*$ as in Equation (3). Finally, the buyer surplus and seller revenue follow directly from the fact that purchase happens at bundle $\boldsymbol{x}^*$ with payment $p^*$. These conclude the proof of Theorem 1.

## D Omitted Proofs in Section 5

### D.1 Proof of Theorem 2

**Theorem 2.** *When $c$ is convex and differentiable, the piece-wise linear concave value function $u^*(\boldsymbol{x})$ defined by Equation (3), with $p^*=\boldsymbol{x}^*\cdot\nabla c(\boldsymbol{x}^*)$ and $\boldsymbol{x}^*=\arg\max_{\boldsymbol{x}\in X}\left[v(\boldsymbol{x})-\boldsymbol{x}\cdot\nabla c(\boldsymbol{x})\right]$, is an optimal buyer imitative value function.*

*Under $u^*(\boldsymbol{x})$, the trade happens at bundle $\boldsymbol{x}^*$ with payment $p^*=\boldsymbol{x}^*\cdot\nabla c(\boldsymbol{x}^*)$. The seller revenue $[\boldsymbol{x}^*\cdot\nabla c(\boldsymbol{x}^*)-c(\boldsymbol{x}^*)]$ is precisely the Bregman divergence $D_c(\boldsymbol{0},\boldsymbol{x}^*)$ between $\boldsymbol{0}$ and $\boldsymbol{x}^*$. The buyer surplus is $[v(\boldsymbol{x}^*)-\boldsymbol{x}^*\cdot\nabla c(\boldsymbol{x}^*)]$.*

*Proof.* Suppose $c(\boldsymbol{x})$ is convex and differentiable. Fix any $\boldsymbol{x}=\overline{\boldsymbol{x}}$. Consider the function $c(\alpha\overline{\boldsymbol{x}})$ with variable $\alpha\in[0,1)$. This is a one-dimensional convex non-decreasing function. Due to convexity, the supremum of $\frac{c(\boldsymbol{x})-c(\alpha\boldsymbol{x})}{1-\alpha}$ over $\alpha\in[0,1)$ equals precisely the derivative of $c(\alpha\cdot\overline{\boldsymbol{x}})$ at $\alpha=1$, which is $\boldsymbol{x}\cdot\nabla c(\boldsymbol{x})$. To find the $\overline{\boldsymbol{x}}$ that maximizes the buyer's revenue, the buyer will pick $\boldsymbol{x}^*=\arg\max_{\boldsymbol{x}\in X}\left[v(\boldsymbol{x})-\boldsymbol{x}\cdot\nabla c(\boldsymbol{x})\right]$. Given the above characterization, the theorem conclusion follows from Theorem 1. $\qquad\square$

## D.2 Proof of Theorem 3

**Theorem 3.** *When $c(\boldsymbol{x})$ is concave, the piece-wise linear concave value function $u^*(\boldsymbol{x})$ defined by Equation (3), with $p^* = c(\boldsymbol{x}^*)$ and $\boldsymbol{x}^* = \arg\max_{\boldsymbol{x} \in X} [v(\boldsymbol{x}) - c(\boldsymbol{x})]$, is an optimal buyer imitative value function.*

*Under $u^*(\boldsymbol{x})$, the trade happens at bundle $\boldsymbol{x}^*$ with payment $p^* = c(\boldsymbol{x}^*)$. The seller revenue will be $0$. The buyer extracts the maximum possible surplus $\max_{\boldsymbol{x} \in X}[v(\boldsymbol{x}) - c(\boldsymbol{x})]$.*

*Proof.* Suppose $c(\boldsymbol{x})$ is concave and differentiable. Fix any $\boldsymbol{x} = \overline{\boldsymbol{x}}$. Consider the function $c(\alpha\overline{\boldsymbol{x}})$ with variable $\alpha \in [0, 1)$. This is a one-dimensional concave non-decreasing function. Due to concavity, the supremum of $\frac{c(\boldsymbol{x}) - c(\alpha\boldsymbol{x})}{1 - \alpha}$ over $\alpha \in [0, 1)$ is achieved at $\alpha = 0$. The supremum thus equals precisely $\frac{c(\boldsymbol{x}) - c(0 \cdot \boldsymbol{x})}{1 - 0} = c(\boldsymbol{x})$. To find the $\overline{\boldsymbol{x}}$ that maximizes the buyer's revenue, the buyer will pick $\boldsymbol{x}^* = \arg\max_{\boldsymbol{x} \in X} [v(\boldsymbol{x}) - c(\boldsymbol{x})]$. Given the above characterization, the theorem conclusion follows from Theorem 1. Specifically, the buyer payment $p^* = c(\boldsymbol{x}^*)$, leading to seller revenue $0$. $\square$

## D.3 Proof of Theorem 4

**Theorem 4.** *[Intractability of Equilibrium] It is NP-hard to approximate the buyer equilibrium surplus in PADD games to be within ratio $1/d^{1-\epsilon}$ for any $\epsilon > 0$. This hardness result holds even when the production cost function $c(\boldsymbol{x})$ is concave and the buyer's true value function $v(\boldsymbol{x})$ is simply the linear function $\sum_{i=1}^{d} x_i$.*

*Proof.* As stated in the theorem, we consider the case where the buyer's true value function is $v(\boldsymbol{x}) = \sum_{i=1}^{d} x_i$ and the seller's production cost function $c(\boldsymbol{x})$ is a concave function that we will construct. Theorem 3 shows in this case, the buyer's optimal surplus is $\max_{\boldsymbol{x} \in X}[\sum_{i=1}^{d} x_i - c(\boldsymbol{x})]$.

Next, we show that this optimization problem is NP-hard to be approximated within any meaningful ratio, as described by the theorem. Our reduction is from the independent set problem. For any connected graph $G = (V, E)$ with $d$ nodes, let node set $V = [d] = \{1, 2, \cdots, d\}$. A set $I$ is an independent set of $G$ if and only if any $i, j \in I$ are not adjacent in $G$. The problem of finding the largest independent set problem is NP-hard, and cannot be approximated within ratio $1/d^{1-\epsilon}$ for any constant $\epsilon > 0$.

Given any instance graph $G = (V, E)$ of the Independent set problem, we construct the following concave production cost function:

$$c(\boldsymbol{x}) = \sum_{i=1}^{d} \min(\sum_{j=1}^{d} a_{ji} x_j, x_i), \text{ where } a_{ji} = 1 \text{ if } (j, i) \in E \text{ and } a_{ji} = 0 \text{ otherwise.}$$

Moreover, the set of feasible bundles is $X = [0, 1]^d$. Note that $c(\boldsymbol{x})$ is a concave function because the minimum of two linear functions is concave and the sum of concave functions remains concave. Moreover, $c(\boldsymbol{x})$ is monotone non-decreasing and $c(\boldsymbol{0}) = 0$. So $c(\boldsymbol{x})$ is indeed a valid cost function for our setting. Under this construction, the maximum possible buyer surplus is the optimal objective of the following optimization problem (OP):

$$\max_{\boldsymbol{x} \in [0,1]^d} U(\boldsymbol{x}) = \sum_{i=1}^{d} x_i - c(\boldsymbol{x}) = \sum_{i=1}^{d} \left[ x_i - \min(\sum_{j=1}^{d} x_j a_{ji}, x_i) \right]. \tag{15}$$

We now show via a reduction from the largest independent set problem that it is NP-hard to approximate OP (15) to be within ratio $1/d^{1-\epsilon}$ for any $\epsilon > 0$. Let $I \subseteq V$ be the maximum independent set. We claim that the optimal objective value of the above optimization problem equals precisely $|I|$, the size of the maximum independent set. For convenience, let term $U_i(\boldsymbol{x}) = x_i - \min(\sum_{j=1}^{d} x_j a_{ji}, x_i)$ and thus $U(\boldsymbol{x}) = \sum_{i=1}^{d} U_i(\boldsymbol{x})$. Note that $U_i(\boldsymbol{x}) \leq 1$ for any $i \in V$.

First, we show that the optimal objective value of OP (15) is at least $|I|$. To see this, consider $\overline{\boldsymbol{x}}$ such that $\overline{x}_i = 1$ if $i \in I$ and $\overline{x}_i = 0$ if $i \notin I$. We argue that for any $i \in I$, $U_i(\overline{\boldsymbol{x}}) = 1$. This is because $\overline{x}_j a_{ji} = 0$ for any $j$ — for any $\overline{x}_j = 1$, we must have $a_{ji} = 0$ as the two nodes $i, j$ are

both in the independent set and thus cannot have an edge between them (i.e., $a_{ji} = 0$). Therefore, $\sum_{j=1}^{d} \overline{x}_j a_{ji} = 0$ and thus $\min(\sum_{j=1}^{d} \overline{x}_j a_{ji}, \overline{x}_i) = 0$. As a consequence, for any $i \in I$, $U_i(\overline{x}) = 1$ and thus the objective of (15) at $\overline{x}$ is at least $|I|$.

Next, we show the reverse direction, i.e., $\max_{x \in [0,1]^d} U(x)$ is at most $|I|$. Note that $U(x)$ is a convex function. So it must achieve the maximum at some vertex $x^*$ of the feasible region, which is a binary vector. Let $S^* \subseteq [d]$ denote the set of the indexes of non-zero values in $x^*$. First of all, for any $i \notin S^*$, $U_i(x^*) \leq x_i^* = 0$. Second, for any $i \in S^*$, if there exists $j \in S^*$ such that $(i, j) \in E$ is an edge, then $x_j^* a_{ji} = 1$ and thus $\min(\sum_{j=1}^{d} x_j^* a_{ji}, x_i^*) = 1$. This implies $U_i(x^*) = 0$. Similarly, $U_j(x^*) = 0$ as well. Finally, for any $i \in S^*$ without any neighbor included in $S^*$, it is easy to see that $U_i(x^*) = 1$. To sum up, only the node $i \in S^*$ that does not have any neighbor included in $S^*$ can have $U_i(x^*) = 1$ whereas any other node $i$ has $U_i(x^*) = 0$. Therefore, $U(x^*)$ is at most the size of the number of independent nodes in $S^*$, which is at most the size of the maximum independent set for $G$, as desired.

So far we have shown that the optimal objective value of OP (15) equals precisely the size of the maximum independent set of $G$. However, we are not done yet to prove the inapproximability of maximizing $U(x)$. This is because $U(x)$ takes fractional variables as input. The fact that it is hard to find an independent set to approximate the size of the maximum independent set does not imply the hardness of finding a fractional variable $x \in X$ to approximate $\max U(x)$.

To prove the inapproximability of the continuous OP (15), we show that any $\alpha$-approximation to OP (15) can be efficiently turned into an $\alpha$-approximation to the largest independent set problem, using ideas from de-randomization. Since it is NP-hard to approximate the largest independent set problem to be within $1/d^{1-\epsilon}$ for any $\epsilon > 0$, this will conclude the proof of the proposition.

Specifically, let $\overline{x} \in [0,1]^d$ be any $\alpha$-approximation to OP (15). We construct a *random* binary vector $\overline{X}$ as follows: $\mathbf{Pr}(\overline{X}_i = 1) = \overline{x}_i$ and $\mathbf{Pr}(\overline{X}_i = 0) = 1 - \overline{x}_i$ for each $i$ independently. By convexity of $U(x)$, we have $\mathbf{E}[U(\overline{X})] \geq U(\mathbf{E}[\overline{X}]) = U(\overline{x})$. In other words, if we pick the random solution $\overline{X}$, the expected objective is at least $U(\overline{x})$. By a standard de-randomization procedure (up to an additive $\epsilon$ difference due to Monte-Carlo sampling),[10] we can efficiently find a binary vector $\overline{x}'$ whose value is also at least $U(\overline{x})$. By a similar argument above, we know that all the independent nodes in $\overline{x}'$ form an independent set whose size is at least $U(\overline{x})$, as desired. $\square$

# E  Omitted Proofs in Section 6

## E.1  Proof of Lemma 2

**Lemma 2.** *The following concave function is optimal to FOP* (6)*:*

$$\overline{u}(x) = \overline{p} \cdot \min\{\frac{x_1}{\overline{x}_1}, \cdots \frac{x_d}{\overline{x}_d}, 1\}, \quad \text{where } \overline{p} = \sup_{\alpha \in [0,1)} \frac{c(\overline{x}) - c(\alpha\overline{x})}{1 - \alpha}. \tag{7}$$

*Proof of Lemma 2.* We first prove the a function of the following format, parameterized by variable $\overline{p}$, is optimal to FOP (6):

$$\overline{u}(x) = \overline{p} \cdot \min\{\frac{x_1}{\overline{x}_1}, \cdots \frac{x_d}{\overline{x}_d}, 1\} \tag{16}$$

To prove this, let $u^*$ be any optimal solution to (6). We now construct another concave value function $\overline{u}(x)$ as follows:

$$\overline{u}(x) = u^*(\overline{x}) \cdot \min\{\frac{x_1}{\overline{x}_1}, \cdots \frac{x_d}{\overline{x}_d}, 1\} \tag{17}$$

---

[10]Specifically, $\mathbf{E}_{\overline{X}}[U(\overline{X})] = \overline{x}_i \mathbf{E}_{\overline{X}}[U(\overline{X}|\overline{X}_i = 1)] + (1 - \overline{x}_i) \mathbf{E}_{\overline{X}}[U(\overline{X}|\overline{X}_i = 0)]$ for each $i$. To de-randomize, we simply calculate $\mathbf{E}_{\overline{X}}[U(\overline{X}|\overline{X}_i = 1)]$ and $\mathbf{E}_{\overline{X}}[U(\overline{X}|\overline{X}_i = 0)]$ through Monte-Carlo sampling, and then pick the larger one.

Next, we first argue that the constructed $\overline{u}$ is still feasible to FOP (6). First, for any $\boldsymbol{x}' = \alpha\overline{\boldsymbol{x}}$ where $\alpha \in [0,1]$, we have

$$\overline{u}(\overline{\boldsymbol{x}}) - c(\overline{\boldsymbol{x}}) = u^*(\overline{\boldsymbol{x}}) - c(\overline{\boldsymbol{x}}) \qquad \text{(by definition of } \overline{u}) \qquad (18)$$
$$\geq u^*(\boldsymbol{x}') - c(\boldsymbol{x}') \qquad \text{(by feasibility of } u^* ) \qquad (19)$$
$$\geq \alpha \cdot u^*(\overline{\boldsymbol{x}}) - c(\boldsymbol{x}') \qquad \text{(By concavity of } u^*, \text{ and } \boldsymbol{x}' = \alpha\overline{\boldsymbol{x}}) \qquad (20)$$
$$= \overline{u}(\boldsymbol{x}') - c(\boldsymbol{x}'). \qquad \text{(by definition of } \overline{u}) \qquad (21)$$

Note the last inequality is by concavity of $u^*$, we have $(1 - \alpha) \cdot u^*(\boldsymbol{0}) + \alpha \cdot u^*(\overline{\boldsymbol{x}}) \leq u^*(\alpha\overline{\boldsymbol{x}})$ where we have $u^*(\boldsymbol{0}) = 0$. Thus, we have $\alpha \cdot u^*(\overline{\boldsymbol{x}}) \leq u^*(\alpha\overline{\boldsymbol{x}}) = u^*(\boldsymbol{x}')$.

Then we consider the case $\boldsymbol{x}' \neq \alpha\overline{\boldsymbol{x}}$. There will be two possible situations if $\boldsymbol{x}' \neq \alpha\overline{\boldsymbol{x}}$:

1. If $\boldsymbol{x}'$ is element-wise greater than $\overline{\boldsymbol{x}}$, then $\overline{u}(\boldsymbol{x}') = u^*(\overline{\boldsymbol{x}}) \cdot \min\{\frac{x'_1}{\overline{x}_1}, \cdots \frac{x'_d}{\overline{x}_d}, 1\} = \overline{u}(\overline{\boldsymbol{x}})$. Thus, we have $\overline{u}(\overline{\boldsymbol{x}}) - c(\overline{\boldsymbol{x}}) \geq \overline{u}(\boldsymbol{x}') - c(\boldsymbol{x}')$ by the monotonicity of $c(\boldsymbol{x})$.

2. If $\boldsymbol{x}'$ is *not* element-wise greater than $\overline{\boldsymbol{x}}$, let $i^* = \operatorname{argmin}_{\{i|x'_i \leq \overline{x}_i\}} \frac{x'_i}{\overline{x}_i}$ and $\alpha^* = \min_{\{i|x'_i \leq \overline{x}_i\}} \frac{x'_i}{\overline{x}_i}$. Then we have $\overline{u}(\boldsymbol{x}') = u^*(\overline{\boldsymbol{x}}) \cdot \min\{\frac{x'_1}{\overline{x}_1}, \cdots \frac{x'_d}{\overline{x}_d}, 1\} = \alpha^* u^*(\overline{\boldsymbol{x}})$. In addition, denote $\widehat{\boldsymbol{x}}' = \alpha^*\overline{\boldsymbol{x}}$. Note that for $\widehat{\boldsymbol{x}}'$, we also have $\overline{u}(\widehat{\boldsymbol{x}}') = u^*(\overline{\boldsymbol{x}}) \cdot \min\{\frac{\widehat{x}'_1}{\overline{x}_1}, \cdots \frac{\widehat{x}'_d}{\overline{x}_d}, 1\} = \alpha^* u^*(\overline{\boldsymbol{x}}) = \overline{u}(\boldsymbol{x}')$. However, we have $c(\widehat{\boldsymbol{x}}') \leq c(\boldsymbol{x}')$ because $\widehat{\boldsymbol{x}}'$ is element-wise less than or equal to $\boldsymbol{x}'$. Thus, we have $\overline{u}(\widehat{\boldsymbol{x}}') - c(\widehat{\boldsymbol{x}}') \geq \overline{u}(\boldsymbol{x}') - c(\boldsymbol{x}')$ by monotonicity of $c(\boldsymbol{x})$. By equations (18)-(21), we have $\overline{u}(\overline{\boldsymbol{x}}) - c(\overline{\boldsymbol{x}}) \geq \overline{u}(\alpha\overline{\boldsymbol{x}}) - c(\alpha\overline{\boldsymbol{x}}), \forall \alpha \in [0,1]$. On the other hand, for any $\boldsymbol{x}' \neq \alpha\overline{\boldsymbol{x}}$ which is not element-wise greater than $\overline{\boldsymbol{x}}$, there must exist a $\widehat{\boldsymbol{x}}' = \alpha^*\overline{\boldsymbol{x}}$ where $\alpha^* = \min_{\{i|x'_i \leq \overline{x}_i\}} \frac{x'_i}{\overline{x}_i} \in [0,1]$ such that $\overline{u}(\boldsymbol{x}') - c(\boldsymbol{x}') \leq \overline{u}(\widehat{\boldsymbol{x}}') - c(\widehat{\boldsymbol{x}}') \leq \overline{u}(\overline{\boldsymbol{x}}) - c(\overline{\boldsymbol{x}})$.

As a result, the constructed $\overline{u}(\boldsymbol{x})$ is feasible to FOP (6) because $\overline{u}(\overline{\boldsymbol{x}}) - c(\overline{\boldsymbol{x}}) \geq \overline{u}(\boldsymbol{x}') - c(\boldsymbol{x}')$ for any $\boldsymbol{x}' \in X$.

Next, we argue that $\overline{u}$ achieves the same buyer utility, and thus must also be optimal. This is because: (1) the feasiblity of $\overline{u}$ implies that the optimal price will be $\overline{u}(\overline{\boldsymbol{x}}) = u^*(\overline{\boldsymbol{x}})$; (2) the optimal buyer amount will then be $\overline{\boldsymbol{x}}$ by breaking ties rule. As a result, buyer achieves the same utility when using $\overline{u}$ and $u^*$, yielding the optimality of $\overline{u}$.

So far we showed that there always exists an optimal $\overline{u}$ of Form (16), which is parameterized $\overline{p} \in \mathbb{R}_{\geq 0}$. We then return to the situation of linear pricing since a linear pricing scheme with unit price $\left(\lambda_1 \cdot \frac{\overline{p}}{\overline{x}_1}, \lambda_2 \cdot \frac{\overline{p}}{\overline{x}_2}, \cdots, \lambda_d \cdot \frac{\overline{p}}{\overline{x}_d}\right)$ where $\sum_i \lambda_i = 1$ is optimal for such a $\overline{u}$ among all concave pricing schemes. The characterization then follows from Theorem 1. $\qquad\square$

## E.2 Proof of Proposition 1

**Proposition 1.** *For the instance in Example 1, there exists pricing scheme class $\mathcal{P}$ with $\mathcal{P}_L \subset \mathcal{P} \subset \mathcal{P}_C$ such that when the seller changes from linear pricing class $\mathcal{P}_L$ to the richer class $\mathcal{P}$, the seller's revenue strictly* decreases *and the buyer's surplus strictly* increases *at the equilibrium of* PADD.

*Proof.* We first analyze the situation of linear pricing. By the characterization in Theorem 2, we know that the optimal purchase bundle satisfies $x^* = \operatorname{argmax}_{\mathbf{x} \in X}[v(\mathbf{x}) - \mathbf{x} \cdot \nabla c(\mathbf{x})]$. Given $v(x)$ and $c(x)$ in Example 1, this solves for $x^* = 0.81$ in the linear pricing setting. Furthermore, this gives $p^* = x^* \cdot \nabla c(x^*) = 1.3122$ and optimal imitative value function $u^* = 1.3122 \cdot \min\{x/0.81, 1\}$. The seller revenue in this case is $p^* - c(x^*) = 0.6561$.

Next, we show that the seller's revenue will strictly decrease at equilibrium when using pricing class $\mathcal{P} = \mathcal{P}_L \cup \{\widehat{p}\}$ that augments linear pricing class $\mathcal{P}_L$ with the following additional choice of a concave pricing function

$$\widetilde{p}(x) = \min(\sqrt{x}, \frac{5}{9}x + \frac{9}{20} - \epsilon)$$

where $\epsilon = 0.05$ is a small constant.

Consider the imitative value function $u^*(x) = \sqrt{x}$ and a particular linear pricing response $p > 0$. In this case, the buyer will purchase an amount $x' = \text{argmax}_{x \in X}[u^*(x) - p \cdot x]$, implying $\frac{1}{2\sqrt{x}} = \frac{du^*(x')}{dx} = p$. Solving for $x'$ gives $x' = \frac{1}{4p^2}$. Now, we solve for the linear pricing response $p_L$ that maximizes the seller revenue.

$$
\begin{aligned}
p_L &= \underset{p \in L}{\text{argmax}}[p \cdot x' - c(x')] \\
&= \underset{p \in L}{\text{argmax}}[p \cdot \frac{1}{4p^2} - \frac{1}{(4p^2)^2}] \\
&= \underset{p \in L}{\text{argmax}}[\frac{1}{4p} - \frac{1}{16p^4}] \\
&= 1
\end{aligned}
$$

Thus, the optimal linear pricing response to $u^*(x) = \sqrt{x}$ is $p_L = 1$. The buyer will purchase $x' = 0.25$, giving the seller a revenue of $1 \cdot 0.25 - 0.25^2 = 0.1875$. Finally, consider the pricing function $\widetilde{p}(x)$ for the buyer's imitative value function $u^*(x) = \sqrt{x}$. We observe that $\frac{d}{dx}(\frac{5}{9}x + \frac{9}{20} - \epsilon) = \frac{5}{9} = \frac{1}{2\sqrt{0.81}} = \frac{d}{dx}(u^*(0.81))$, meaning $\text{argmax}_{x \in X}[u^*(x) - p^*(x)] = 0.81$. Thus, the buyer can purchase $x = 0.81$ and the seller will get revenue $0.2439 - \epsilon = 0.1939$. This is strictly larger than the seller's revenue $0.1875$ from the optimal linear pricing scheme. Thus the seller's optimal pricing scheme from $\mathcal{P}$ is $\widetilde{p}(x)$.

Finally, note that $\text{argmax}_{x \in X}[v(x) - \widetilde{p}(x)] = 0.81$, meaning the optimal bundle for the buyer to purchase when the seller responds with $\widetilde{p}$ is $x = 0.81$. In this case the buyer surplus is $v(0.81) - \widetilde{p}(0.81) = 8.1 - 0.9 + \epsilon > 8.1 - 1.3122 = v(0.81) - p_L \cdot 0.81$, meaning the optimal true buyer surplus given pricing function $\widetilde{p}(x)$ is greater than the optimal true buyer surplus given any linear pricing response. Thus, $u^*(x)$ is an optimal imitative function under pricing class $\mathcal{P}$ and the seller revenue of $0.2439 - \epsilon = 0.1939$ is strictly lower than the seller revenue of $0.6561$ under linear pricing class $\mathcal{P}_L$. □