# OpenReview forum: "The Limits of Optimal Pricing in the Dark"
_NeurIPS.cc/2021/Conference — NeurIPS 2021 Poster_

### Official Review · Reviewer_Abq4 · 2021-07-14

**Rating:** 5
**Confidence:** 3

**Summary:**

The paper discusses an economic strategic learning scenario, where a seller and a buyer repeatedly engage in selling a bundle of d divisible goods, and where the seller attempt to learn the pricing scheme leading to a maximal revenue.

In this scenario, there is only one seller and one buyer, i.e. no competition could be introduced (such as what would happen in an auction for example). The buyer could manipulate the seller by refusing to buy at certain prices even if that single purchase has a positive utility to them, so as to lead the seller to change the prices in a way that improves the buyer's utility in the future.

The proposed scheme is a simple Stackelberg model, where the buyer commits to "pretending" they have a different utility function (i.e acting as if they were making local buying decisions without trying to manipulate under a utility that is different to their true one).

The authors characterize teh the optimal value function the buyer should imitate, and examines the resulting equilibrium. Under concave production functions, the seller ends up with zero revenue (i.e. the buyer would manipulate in a way that extracts all the surplus), whereas for a convex production they give an equilibrium with a positive seller utility. Another key result is that a different class of pricing schemes does not mitigage this effect, and would in fact be harmful for the seller.

**Limitations And Societal Impact:**

The authors fully discuss the limitations of their work, and I do not see a need to dig deeper into possible societal impact.

**Main Review:**

The problem studied here seems to be a natural economic problem, and the results are insightful.

Economically, the scenario strikes me as a bit odd, as there is no competition here (so alternatively, one might view this as a seller and buyer negotiating a pricing scheme for multiple transactions in the future, in a case where the seller has no knowledge of the buyer's utility). This is in sharp contrast to perfect competition between firms or a repeated auction. However, the authors provide some justification as to why some domains might behave this way.

In terms of the key takeaways, I think they relate to economics rather than machine learning. While some of the related work deals with e.g. classification in strategic settings, I feel like the results here have much less to do with learning, but rather with optimization and economics, so I am not sure whether this is the best venue for this work (as opposed to say ACM EC, WINE or even some of the tracks at WWW).

The analysis is non-trivial, but again I'm not sure whether most in the general ML community would be able to put the techniques applied here to use in their own domains. Overall, the authors need to do a better job at tying this to related ML literature (e.g. a table comparing the results here with strategic classification). I also think a comparison with other Stackelberg domains is in order; the key insight here is that a very simple manipulation works --- the buyer just needs to pretend they have a different valuation function and follow that in a naive (truthful) way, so a more sophisticated manipulation is not required (in contrast to other Stackleberg domains). What are the connections between this and the field of mechanism design?

To conclude, interesting paper, but not a great fit for the conference - not sure what the key takeaway message is for an ML audience. Also, the authors need to better connect it to existing work. On the positive side, interesting technique, and a good presentation of the key results with some insight.

**Time Spent Reviewing:**

5 hours

---

> ### Author Response · Authors · 2021-08-08
> **Response to Reviewer Abq4**
>
> We thank the reviewer for the insightful comments and for appreciating our results. The reviewer's major concern is whether the paper is a good fit for NeurIPS. We have no doubt about this at all, and the following are some of the reasons.
> - First, regarding relevance to the ML community, our major motivation directly comes from recent machine learning literature of learning prices from honest buyer responses [3, 8, 32, 38]. Our work tries to understand the limits of these previously developed learning algorithms when they are used against a strategic buyer who may manipulate the seller's learning. From another perspective, one can also view our results as designing an effective "attack" strategy for these learning algorithms. Both of the studies of learning in the presence of strategic behaviors and attacks to ML algorithms have attracted significant recent interest in AI and ML, and our work subscribes to this literature.
> - Second, even the topic of "algorithmic game theory'' is listed as one of the targeting areas in the NeurIPS'21 call for papers website. Specifically,   the "Theory'' bullets list three areas as examples:  control theory, learning theory, algorithmic game theory.
> - Finally, many of our cited papers on algorithmic game theory have been accepted to previous years of NeurIPS (e.g. [4, 5, 37]). Even beyond NeurIPS, the AISTATS 2020 paper *Robust Stackelberg buyers in repeated auctions* suggested by Reviewer-ocDy is also on a similar topic as us: it studies buyer’s simple and robust strategies against a revenue-maximizing seller.

---

> > ### Author Response · Authors · 2021-08-26
> > **Follow-up Discussion**
> >
> > Dear Reviewer Abq4,
> >
> > Thank you again for your time and for appreciating our results. We believe we have responded properly to your major concern, however please let us know if you might have any additional questions. We would be happy to have a discussion.
> >
> > Sincerely, The authors

---

### Official Review · Reviewer_ocDy · 2021-07-15

**Rating:** 7
**Confidence:** 3

**Summary:**

This paper consider a problem of buying/selling multiple divisible items with a posted price mechanism between one buyer and one seller, in a noiseless setting. It consider the problem that through repeated trades, buyer and seller learn from each other, leading them to adopt strategies that maximizes their revenue over time, breaking classical incentive compatibility properties. More precisely, this paper considers that
* given a mechanism (seller) and a value function (buyer), the buyer plays a one-shot best response.
* given a value function (buyer), the seller chooses the optimal mechanism (knowing the buyer will play best response)
* given this previous knowledge, the buyer chooses a *fake* value function to announce to the seller to optimize his objective.

This paper describes the properties of the solutions of this tri-level optimization problem and characterize the complexity of solving it.

**Ethical Concerns:**

Not that I can think of.

**Limitations And Societal Impact:**

see main review

**Main Review:**

**Clarity and quality**
* The paper is easy to read and maths are clearly written. I just wish the appendix was more self-contained, e.g. with cited results and optionally their proofs restated, especially for intermediate results that are pointing towards articles behind paywalls and thus not available.
* This paper manages to bring some good insights on the complex problem of understanding buyers strategies and more especially how seller optimization impacts buyers' strategies.
* Maybe a more recent work to position in you related work (related to Tang and Zheng 2018) *Robust Stackelberg buyers in repeated auctions.* (AISTATS 2020).


**Originality and significance**
* While there is a significant amount of work about how seller should behave in repeated settings, there is much little work on the strategies buyers could use. As such, I find this paper original.
* Some comments about the setting and maybe an ask for better explaining its limitations:
  * It studies an equilibrium of a repeated game, but does not provide insights on whether this equilibrium is reachable by dynamic strategies.
  * It considers a specific class of strategies for the buyer: a one shot best-response $x^*(.)$ based on a fake value function $u$ concave and monotone, non-decreasing. I'm wondering whether when optimizing on *any* fake value function $u$ (not necessarily concave), the optimum $u^*$ is still concave when $v$ is so or not. For instance, in Tang and Zheng 2018, even if the true value distribution is regular, the optimal fake distribution is not regular. Is there a similar phenomenon here ?
* Maybe a future direction: the seller generally does not have a perfect knowledge of $u$ (but estimates it from historical interactions for instance), how sensitive is the optimal $u$ to the seller only knowing it at precision $\varepsilon$ ?

**Time Spent Reviewing:**

6

---

> ### Author Response · Authors · 2021-08-08
> **Response to Reviewer ocDy**
>
> We are excited to hear the reviewer's encouraging feedback and sincerely appreciate your time. We will implement the reviewer's suggestions about the very recent related work and make the appendix more self-contained. We fully agree with the intriguing future direction suggested by the reviewer (see also line 399 in our conclusion section). Below we clarify two additional points.
> - For the reviewer’s comment about whether the equilibrium is reachable, the answer is *yes*, as long as the seller uses a learning algorithm that can effectively learn the optimal pricing against an honestly responding buyer.
> - For the question about the assumption of concave buyer value function $v(\cdot)$, this is a great question. Allowing general $v$ turns out to be extremely challenging since many nice properties of the seller's optimal pricing problem are lost (in fact, even the seller's optimal pricing problem may become intractable). This is perhaps one of the key reasons that almost all previous works about optimal pricing [6, 8, 32, 38], including standard textbooks (e.g., [MWG’95 Microeconomic Theory]), adopt concave buyer value function assumptions. Our work inherited this assumption from the standard literature.  Alternatively, one can also interpret concave buyer value as the seller's prior knowledge. It is an intriguing open problem to consider a seller without this prior knowledge.

---

### Official Review · Reviewer_dqi7 · 2021-07-16

**Rating:** 8
**Confidence:** 3

**Summary:**

This paper considers the dynamic of a pricing game between a strategic buyer and a revenue-maximizing seller. The main contribution of this paper is that they give a complete characterization of buyer's optimal strategy and seller's corresponding response (which forms an equilibrium).



**Limitations And Societal Impact:**

The authors have adequately addressed the limitations and I see no potential negative societal impact of this work.

**Main Review:**

This paper considers the dynamic of a pricing game. The story of this paper is like, in reality, sellers often have no chance to gather any information about the potential buyer. Therefore the only way to know something about the preference is learning buyer's valuation function through repeated auctions. However the buyer could play strategically to improve his own utility. This paper considers a quite natural class of strategies of the buyer -- he just pretends their valuation to be another one and plays consistently with this fake valuation. Thus, the only thing that the seller can do is finding the optimal to maximize her revenue. This paper studies the best responses for each player in this game when seller's pricing is restricted to linear pricing.

The main contribution is that this paper fully characterizes the equilibrium of the game. It is a complex tri-level functional optimization problem, and the result equilibrium is surprisingly neat.

Overall I think it's a paper of good quality. The story is interesting to me and setting just makes sense. The highlight of this work is that the characterization of linear pricing is complete and neat. I also really like discussion of non-linear pricing schemes and insightful observations of the possibility of over-exploiting buyer's incentives. I would encourage the author to make proof sketches not that dense, although much of abstraction comes from the nature of functional optimization, maybe adding some examples would help.

**Time Spent Reviewing:**

4

---

> ### Author Response · Authors · 2021-08-08
> **Response to Reviewer dqi7**
>
> We thank the reviewer for the encouraging feedback and for appreciating our results. We will implement the suggested writing comments in the final version.

---

### Official Review · Reviewer_ZMGR · 2021-07-18

**Rating:** 6
**Confidence:** 3

**Summary:**

This paper studies the problem of pricing against a strategic buyer in the digital good market. The authors consider a two-player game environment in which the buyer first commits to imitate a different value function and react optimally according to the imitative value function. And the seller optimizes the pricing strategy against the imitative value function.

The authors fully characterize the optimal imitative function for the buyer as well as seller's revenue and buyer's surplus in the equilibrium when the seller adopts linear pricing. The results demonstrate a sharp contrast between concave production cost and convex production cost. In particular, for concave production cost, the seller obtains 0 revenue. The authors also provide results for pricing schemes beyond linear pricing.

**Limitations And Societal Impact:**

The model choice, in which the buyer commits first and then the seller responds, needs more justification.

**Main Review:**

This paper is generally well-written, clear, and easy to follow. The collection of results seems to be novel and technically strong. The sharp contrast between convex and concave production cost is very interesting and provides useful insights for practice. The reviewer did not check all the proofs but they all look plausible in hindsight.

Comments:

1. If the reviewer understands correctly, although the problem studied in this paper is better motivated in a repeated setting, the concrete setting considered in this paper is essentially a one-shot game instead of a repeated game (e.g., see Definition 1). In other words, the repeated game is essentially compressed into a one-shot game in this paper.

2. It is unclear whether that the buyer commits first and then the seller responds gives the buyer a disadvantage or an advantage. The typical order in online pricing would be that the seller commits to a mechanism first, and then the buyer best responds to the mechanism. In a one-shot game, it would be the seller commits to the allocation and payment rule (depending on the buyer's imitative value function) and then the buyer best responds with an imitative value function. The reviewer is curious about what the results would look like in this setting. In Tang and Zeng [35], if the reviewer understands correctly, the seller already commits to using the Myerson's auction (or the second-price auction) first and then the buyer best responds with a fake distribution.

3. The reviewer appreciates the nice plots in Figure 2. It would be great if the authors can provide high-level intuitions behind Theorem 2.

4. For Theorem 4, the reviewer wonders whether the problem would be tractable when the production cost is convex.

**Time Spent Reviewing:**

2

---

> ### Author Response · Authors · 2021-08-08
> **Response to Reviewer ZMGR**
>
> We thank the reviewer for the insightful comments and for appreciating our results. We will implement the reviewer's suggestion about adding some high-level intuition behind Theorem 2 in our paper. Below we further clarify two points in the reviewer's comments:
> - For comment 2: Our model is actually similar to Tang and Zeng [35]. In our PADD model, before step 1 when the buyer commits to an imitative value function, there is an implicit "step 0'' in which the seller commits to using the optimal pricing scheme w.r.t. the buyer's imitative value function. We thank the reviewer for pointing out this confusing part and will make this point clear in our next version.
> - For comment 4: This is a great question. As noted in Remark 1 and Theorem 2, the optimal $x^*$ under convex cost function $c(x)$ is computed by $\text{argmax}_x [v(x) - x \cdot \nabla c(x)]$. Therefore, whether the problem is tractable  will depend on the specific convex cost function $c(x)$, though intuitively this problem appears to be more tractable than concave $c(x)$.

---

### Decision · Program_Chairs · 2021-09-27

**Decision:**

Accept (Poster)

**Comment:**

This paper deals with a novel novel question, provides a non-trivial and gets to a surprisingly clean result.

There were some concerns regarding the relation to Tang and Zheng and the learning angle to justify publication in NeurIPS, but both points were adequately addressed by the authors in their response.

The review team is confident that this is a strong contribution to NeurIPS.